# Memory-Assisted Sub-Prototype Mining for Universal Domain Adaptation

## Abstract

Universal domain adaptation aims to align the classes and reduce the feature gap between the same category of the source and target domains. The target private category is set as the unknown class during the adaptation process, as it is not included in the source domain. However, most existing methods overlook the intra-class structure within a category, especially in cases where there exists significant concept shift between the samples belonging to the same category. When samples with large concept shift are forced to be pushed together, it may negatively affect the adaptation performance. Moreover, from the interpretability aspect, it is unreasonable to align visual features with significant differences, such as fighter jets and civil aircraft, into the same category. Unfortunately, due to such semantic ambiguity and annotation cost, categories are not always classified in detail, making it difficult for the model to perform precise adaptation. To address these issues, we propose a novel Memory-Assisted Sub-Prototype Mining (MemSPM) method that can learn the differences between samples belonging to the same category and mine sub-classes when there exists significant concept shift between them. By doing so, our model learns a more reasonable feature space that enhances the transferability and reflects the inherent differences among samples annotated as the same category. We evaluate the effectiveness of our MemSPM method over multiple scenarios, including UniDA, OSDA, and PDA. Our method achieves state-of-the-art performance on four benchmarks in most cases.

## 1 Introduction

Unsupervised Domain Adaptation (UDA) [15, 22, 41, 44, 9, 19, 21] has become a crucial research area of transfer learning, as it allows models trained on a specific dataset to be applied to related but distinct domains. However, traditional UDA methods are limited by the assumption that the source and target domains have to share the same label space. This assumption is problematic in real-world scenarios where the target distribution is complex, open, and diverse. Universal Domain Adaptation (UniDA) represents a strategy to address the limitations of traditional unsupervised domain adaptation methods. In the UniDA, the target domain have a different label set than the source domain. The goal is to correctly classify target domain samples belonging to the shared classes in the source label set, while any samples not conforming to the source label set are treated as "unknown". The term "universal" characterizes UniDA as not relying on prior knowledge about the label sets of the target domain. UniDA relaxes the assumption of a shared class space while aims to learn domain-invariant features across a more broad range of domains.

Despite being widely explored, most existing universal domain adaptation methods [24, 47, 40, 39, 6, 34, 8, 26] overlook the internal structure intrinsically presented within each image category. These methods aim to align the common classes between the source and target domains for adaptation, but

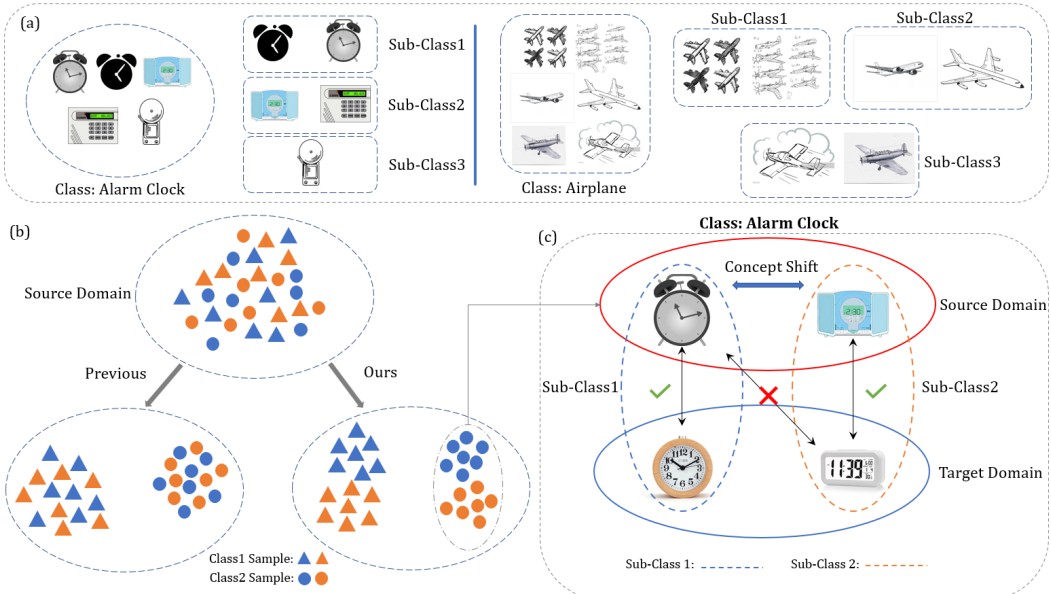

Figure 1: Illustration of our motivation. (a) Examples of concept shift and intra-class diversity in DA benchmarks. For the class of alarm clock, we find that digital clock, pointer clock and alarm bell should be set in different sub-classes. For the class of airplane, we find that images containing more than one plane, single jetliner, and turboprop aircraft should be differently treated for adaptation. (b) Previous methods utilize one-hot labels to guide classifying without considering the intra-class distinction. Consequently, the model forces all samples from the same class to converge towards a single center, disregarding the diversity in the class. Our method clusters samples with large intra-class difference into separate sub-class, providing a more accurate representation. (c) During domain adaptation by our design, the samples in the target domain can also be aligned near the sub-class centers with similar features rather than just the class centers determined by labels.

usually train a model to learn the class "prototype" representing each annotated category. This is particularly controversial when significant concept shift exists between samples belonging to the same category. These differences can lead to sub-optimal feature learning and adaptation if the intra-class structure is neglected during training. Since such kind of semantic ambiguity without fine-grained category labels almost happens in all the DA benchmarks, all the methods will encounter this issue.

In this paper, we aim to propose a method to learn the detailed intra-class distinction and mine "sub-prototypes" for better alignment and adaptation. This kind of sub-prototype is the further subdivision of each category-level prototype, which represents the "sub-class" of the annotated categories. The main idea of our proposed approach lies in its utilization of a learnable memory structure to learn sub-prototypes for their corresponding sub-classes. This can optimize the construction and refinement of the feature space, bolstering the classifier's ability to distinguish class-wise relationships and improve the model's transferability across domains. A comparison between our proposed sub-prototypes mining approach and previous methods is illustrated in Figure 1. In previous methods, samples within a category were forced to be aligned together in the feature space regardless of whether there exist significant differences among them because the labels were one-hot encoded. Contrastively, our sub-prototypes' feature space distinguishes sub-classes with apparent differences within the category, thus improving the model's accuracy of domain adaption and interpretability.

Our proposed approach, named memory-assisted sub-prototype mining (MemSPM), is inspired by the memory mechanism works [17, 10, 45, 36]. In our approach, the memory generates sub-prototypes that embody sub-classes learned from the source domain. During testing of the target samples, the encoder produces embedding that are compared to source domain sub-prototypes learned in the memory. Subsequently, a embedding for the query sample is generated through weighted sub-prototype sampling in the memory. This results in reduced domain shifts before the embedding give into the classifier. Our proposal of sub-prototypes mining, which are learned from the source domain

memory, improves the universal domain adaptation performance by promoting more refined visual concept alignment.

MemSPM approach has been evaluated on four benchmark datasets (Office-31 [37], Office-Home [46], VisDA [33],and Domain-Net [32]), under various category shift scenarios, including PDA, OSDA, and UniDA. Our MemSPM method achieves state-of-the-art performance in most cases. Moreover, we design a visualization module for the sub-prototype learned by our memory to demonstrate the interpretability of MemSPM. Our contributions can be highlighted as follows:

- We study the UniDA problem from a new aspect, which focuses on the negative impacts caused by overlooking the intra-class structure within a category when simply adopting one-hot labels.

- We propose Memory-Assisted Sub-Prototype Mining(MemSPM), which explores the memory mechanism to learn sub-prototypes for improving the model's adaption performance and interpretability. Meanwhile, visualizations reveal the sub-prototypes stored in memory, which demonstrate the interpretability of MemSPM approach.

- Extensive experiments on four benchmarks verify the superior performance of our proposed MemSPM compared with previous works.

## 2  Related Work

**Closed-Set Domain Adaptation (CSDA).** To mitigate the performance degradation caused by the closed-set domain shift, [16, 29, 48] introduce adversarial learning methods with the domain discriminator, aiming to minimize the domain gap between source and target domains. Beyond the use of the additional domain discriminator, some studies [41, 23, 50, 30, 13] have explored the use of two task-specific classifiers, otherwise referred to as bi-classifier, to implicitly achieve the adversarial learning. However, the previously mentioned methods for CSDA cannot be directly applied in scenarios involving the category shift.

**Partial Domain Adaptation (PDA).** PDA posits that private classes are exclusive to the source domain. Representative PDA methods, such as those discussed in [3, 49], employ domain discriminators with weight adjustments or utilize source samples based on their resemblance to the target domain [5]. Methods incorporating residual correction blocks in PDA have been introduced by Li et al. and Liang et al. [25, 27]. Other research [7, 11, 38] explores the use of Reinforcement Learning for source data selection within the context of PDA.

**Open-Set Domain Adaptation (OSDA).** Saito et al. [42] developed a classifier inclusive of an additional 'unknown' class intended to differentiate categories unique to the target domain. Liu et al. [28] and Shermin et al. [43] propose assigning individual weights to each sample depending on their importance during domain adaptation. Jang et al. [20] strive to align the source and target-known distributions, while concurrently distinguishing the target-unknown distribution within the feature alignment process. The above PDA and OSDA methods are limited to specific category shift.

**Universal Domain Adaptation (UniDA)** You et al. [47] proposed Universal Adaptation Network (UAN) to deal with the UniDA setting that the label set of target domain is unknown. Li et al. [24] proposed Domain Consensus Clustering to differentiate the private classes rather than treat the unknow classes as one class. Saito et al. [40] suggested that using the minimum inter-class distance in the source domain as a threshold can be an effective approach for distinguishing between "known" and "unknown" samples in the target domain. However, most existing methods [24, 47, 40, 39, 6, 34, 8, 26] overlook the intra-class distinction within one category, especially in cases where there exists significant concept shift between the samples belonging to the same category.

## 3  Proposed Methods

### 3.1  Preliminaries

In unsupervised domain adaptation, we are provided with labeled source samples $\mathcal{D}^s = \{x_i^s, y_i^s\}_{i=1}^{n^s}$ and unlabeled target samples $\mathcal{D}^t = \{(x_i^t)\}_{i=1}^{n^t}$. As the label set for each domain in UniDA setting may not be identical, we use $C_s$ and $C_t$ to represent label sets for the two domains, respectively.

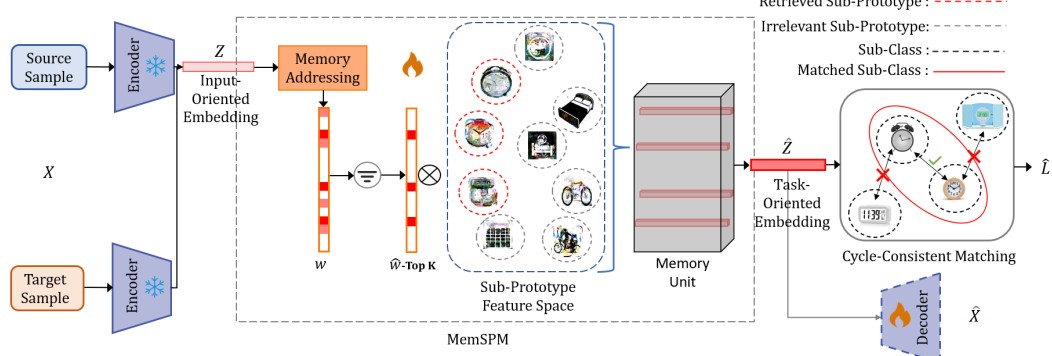

Figure 2: Our model first utilizes a fixed pre-trained model as the encoder to extract input-oriented embedding given an input sample. The extracted input-oriented embedding is then compared with sub-prototypes learned in memory to find the closest $K$. These $K$ are then weighted-averaged into a task-oriented embedding to represent the input, and used for learning downstream tasks. During the UniDA process, we adopt the cycle-consistent matching method on the task-oriented embedding $\hat{Z}$ generated from the memory. Moreover, a decoder is designed to reconstruct the image, allowing for visualizing of the sub-prototypes in memory and verifying of the effectiveness of sub-class learning.

Then, we denote $C = C_s \cap C_t$ as the common label set. $\hat{C}_s$, $\hat{C}_t$ are denoted as the private label sets of the source domain and target domain, respectively. We aim to train a model on $\mathcal{D}^s$ and $\mathcal{D}^t$ to classify target samples into $|C| + 1$ classes, where private samples are treated as unknown class.

Our method aims to address the issue of intra-class concept shift that often exists within the labeled categories in most datasets, which is overlooked by previous methods. Our method enables the model to learn an adaptive feature space that better aligns fine-grained sub-class concepts, taking into account the diversity present within each category. Let $X$ denotes the input query, $Z$ denotes the embedding extracted by the encoder, $L$ denotes the data labels, $\hat{Z}$ denotes the embedding obtained from the memory, $\hat{X}$ denotes the visualization of the memory, $\hat{L}$ denotes the prediction of the input query, and the $K$ denotes the top-K relevant sub-prototypes, respectively. The overall pipeline is presented in Figure 2. More details will be described in the following sub-sections.

### 3.2 Input-Oriented Embedding vs. Task-Oriented Embedding

Usually, the image feature extracted by a visual encoder is directly used for learning downstream tasks. We call this kind of feature as input-oriented embedding. However, it heavily relys on the original image content. Since different samples of the same category always varies significantly in their visual features, categorization based on the input-oriented embedding sometimes is unattainable. In our pipeline, we simply adopt a CLIP-based[35] pre-trained visual encoder to extract the input-oriented embeddings, which is not directly used for learning our downstream task.

In our MemSPM, we propose to generate task-oriented embedding, which is obtained by serving input-oriented embedding as a query to retrieve the sub-prototypes from our memory unit. We define $f_{encode}^{fixed}(\cdot) : X \rightarrow Z$ to represent the fixed pre-trained encoder and $f_{class}^{UniDA}(\cdot) : \hat{Z} \rightarrow \hat{L}$ to represent the UniDA classifier. The input-oriented embedding $Z$ is used to retrieve the relevant sub-prototypes from the memory. The task-oriented embedding $\hat{Z}$ is obtained using the retrieved sub-prototypes for classification tasks. In conventional ways, $\hat{Z} = Z$, which means the $\hat{Z}$ is obtained directly from $Z$. Our method obtains the $\hat{Z}$ by retrieving the sub-prototypes from the memory, which differenciates $\hat{Z}$ with $Z$, and eliminates the domain-specific information from the target domain during the testing phase. As a result, it improves the performance of $f_{class}^{UniDA}(\cdot)$ when performing UniDA.

### 3.3 Memory-Assisted Sub-Prototype Mining

The memory module proposed in MemSPM consists of two key components: a memory unit responsible for learning sub-prototypes, and an attention-based addressing [18] operator to obtain better task-oriented representation $\hat{Z}$ for the query, which is more domain-invariant.

### 3.3.1 Memory Structure with Partitioned Sub-Prototype

The memory in MemSPM is represented as a matrix, denoted by $M \in \mathbb{R}^{N \times S \times D}$, where $N$ indicates the number of memory items stored, $S$ refers to the number of sub-prototypes partitioned in each memory item, and $D$ represents the dimension of each sub-prototype. For convenience, we assume $D$ is the same to the dimension of $Z \in \mathbb{R}^C$ ( $\mathbb{R}^D = \mathbb{R}^C$). Let the vector $m_{i,j}, \forall i \in [N]$ denote the $i$-th row of $M$, where $[N]$ denotes the set of integers from 1 to $N$, $\forall j \in [S]$ denote the $j$-th sub-prototype of $M$ items, where $[S]$ denotes the set of integers from 1 to $S$. Each $m_i$ denotes a memory item. Given a embedding $Z \in \mathbb{R}^D$, the memory module obtains $\hat{Z}$ through a soft addressing vector $W \in \mathbb{R}^{1 \times 1 \times N}$ as follows:

$$\hat{Z} = W \cdot M = \Sigma_{i=1}^N w_{i,j=s_i} \cdot m_{i,j=s_i}, \tag{1}$$

$$w_{i,j=s_i} = \operatorname{argmax}(w_{i,j}, dim = 1), \tag{2}$$

where $W$ is a vector with non-negative entries that indicate the max attention weight of each item's sub-prototype, $s_i$ denotes the index of the sub-prototype in the $i$-th item and $w_{i,j=s_i}$ denotes the $i, j = s_i$-th entry of $W$. The hyperparameter $N$ determines the maximum capacity for memory items and the hyper-parameter $S$ defines the number of sub-prototypes in each memory item. The effect of different setting of hyper-parameters is evaluated in Section 4.

### 3.3.2 Sub-Prototype Addressing and Retrieving

In MemSPM, the memory $M$ is designed to learn the sub-prototypes to represent the input-oriented embedding $Z$. We define the memory as a content addressable memory [17, 10, 45, 36] that allows for direct referencing of the content of the memory being matched. The sub-prototype is retrieved by attention weights $W$ which are computed based on the similarity between the sub-prototypes in the memory items and the input-oriented embedding $Z$. To calculate the weight $w_{i,j}$, we use a softmax operation:

$$w_{i,j} = \frac{\exp(d(z, m_{i,j}))}{\Sigma_{n=1}^N \Sigma_{s=1}^S \exp(d(z, m_{n,s}))}, \tag{3}$$

where $d(\cdot, \cdot)$ denotes cosine similarity measurement. As indicated by Eq. 1 and 3, the memory module retrieves the sub-prototype that is most similar to $Z$ from each memory item in order to obtain the new representation embedding $\hat{Z}$. As a consequence of utilizing the adaptive threshold addressing technique(Section 3.3.3), only the $K$ can be utilized to obtain a task-oriented embedding $\hat{Z}$, that serves to represent the encoded embedding $Z$.

### 3.3.3 Adaptive Threshold Technique for More Efficient Memory

Limiting the amount of sub-prototypes retrieved can enhance memory utilization and avoid negative impacts on unrelated sub-prototypes during model parameter updates. Despite the natural reduction in the number of selected memory items, the attention-based addressing mechanism may still lead to the combination of small attention weight items into the output embedding $\hat{Z}$, which have negative impact on the classifier and sub-prototypes in the memory. Therefore, it is necessary to impose a mandatory quantity limit on the amount of the relevant sub-prototypes retrieved. To address this issue, we apply a adaptive threshold operation to restrict the amount of sub-prototypes retrieved in a forward process.

$$\hat{w}_{i,j=s_i} = \begin{cases} w_{i,j=s_i}, & w_{i,j=s_i} > \lambda \\ 0, & \text{other} \end{cases} \tag{4}$$

where $\hat{w}_{i,j=s_i}$ denotes the $i, j = s_i$-th entry of $\hat{w}$, the $\lambda$ denotes the adaptive threshold:

$$\lambda = \operatorname{argmin}(topk(w)). \tag{5}$$

Directly implementing the backward for the discontinuous function in Eq. 4 is not a easy task. For simplicity, we use the method [17]that rewrites the operation using the continuous ReLU activation function as:

$$\hat{w}_{i,j=s_i} = \frac{\max(w_{i,j=s_i} - \lambda) \cdot w_{i,j=s_i}}{|w_{i,j=s_i} - \lambda| + \epsilon}, \tag{6}$$

where $max(\cdot, 0)$ is commonly referred to as the ReLU activation function, and $\epsilon$ is a small positive scalar. The prototype $\hat{Z}$ will be obtained by $\hat{Z} = \hat{W} \cdot M$. The adaptive threshold addressing encourages the model to represent embedding $Z$ using fewer but more relevant sub-prototypes, leading to learning more effective feature in memory and reducing the impact on irrelevant sub-prototypes.

### 3.4 Visualization and Interpretability

We denote $f_{decode}^{unfixed}(\cdot) : \hat{Z} \to \hat{X}$ to represent the decoder. The decoder is trained to visualize what has been learned in the memory by taking the retrieved sub-prototype as input. From an interpretability perspective, each encoded embedding $Z$ calculates the cosine similarity to find the top-$K$ fitting sub-prototype representation for the given input-oriented embedding. Then, these sub-prototypes are combined to represent the $Z$ in $\hat{Z}$. The sub-prototype in this process can be regarded as the visual description for the input embedding $Z$. In other word, the input image is much like the sub-classes represented by these sub-prototypes. In this way, samples with significant intra-class differences will be matched to different sub-prototypes, thereby distinguishing different sub-classes. The use of a reconstruction auxiliary task can visualize the sub-prototypes in memory to confirm whether our approach has learned intra-class differences for the annotated category. The results of this visualization are demonstrated in Figure 3.

### 3.5 Cycle-Consistent Alignment and Adaption

Once the sub-prototypes are mined through memory learning, the method of cycle-consistent matching, inspired by DCC [24], is employed to align the embedding $\hat{Z}$. The cycle-consistent matching is preferred due to it can provides a better fit to the memory structure compared to other UniDA methods. The other method, One-vs-All Network (OVANet), proposed by Saito et al. [40], needs to train the memory multiple times, which can lead to a significant computational overhead. In brief, the Cycle-Consistent Alignment provides a solution by iteratively learning a consensus set of clusters between the two domains. The consensus clusters are identified based on the similarity of the prototypes, which is measured using a similarity metric. The similarity metric is calculated on the feature representations of the prototypes. For unknown classes, we set the size $N$ of our memory during the initial phase to be larger than the number of possible sub-classes that may be learned in the source domain. This size is a hyperparameter that is adjusted based on the dataset size. Redundant sub-prototypes are invoked to represent the $\hat{Z}$, when encountering unknown classes, allowing for an improved distance separation between unknown and known classes in the feature space.

**Training Objective**. The adaptation loss in our training is similar to that of DCC, as $\mathcal{L}_{DA}$:

$$\mathcal{L}_{DA} = \mathcal{L}_{ce} + \lambda_1 \mathcal{L}_{cdd} + \lambda_2 \mathcal{L}_{reg}, \tag{7}$$

where the $\mathcal{L}_{ce}$ denotes the cross-entropy loss on source samples, $\mathcal{L}_{cdd}$ denotes the domain alignment loss and $\mathcal{L}_{reg}$ denotes the regularizer. For the auxiliary reconstruction task, we add a mean-squared-error (MSE) loss function, denoted as $\mathcal{L}_{rec}$. Thus, the model is optimized with:

$$\mathcal{L} = \mathcal{L}_{DA} + \lambda_3 \mathcal{L}_{rec} = \mathcal{L}_{ce} + \lambda_1 \mathcal{L}_{cdd} + \lambda_2 \mathcal{L}_{reg} + \lambda_3 \mathcal{L}_{rec}. \tag{8}$$

## 4 Experiments

### 4.1 Datasets and Evaluation Metrics

We first conduct the experiments in the UniDA setting [47] where private classes exist in both domains. Moreover, we also evaluate our approach on two other sub-cases, namely Open-Set Domain Adaptation (OSDA) and Partial Domain Adaptation (PDA).

**Datasets**. Our experiments are conducted on four datasets: Office-31 [37], which contains 4652 images from three domains (DSLR, Amazon, and Webcam); OfficeHome

Table 2: H-score (%) comparison in UniDA scenario on DomainNet, VisDA and Office-31,some results are cited from [24, 34]

| Method | Backbone | DomainNet | | | | | | | VisDA | Office-31 | | | | | | |
|---|---|---|---|---|---|---|---|---|---|---|---|---|---|---|---|---|
| | | P2R | P2S | R2P | R2S | S2P | S2R | Avg | S2R | A2D | A2W | D2A | D2W | W2A | W2D | Avg |
| UAN [47] | | 41.9 | 39.1 | 43.6 | 38.7 | 38.9 | 43.7 | 41.0 | 34.8 | 59.7 | 58.6 | 60.1 | 70.6 | 60.3 | 71.4 | 63.5 |
| CMU [14] | | 50.8 | 45.1 | 52.2 | 45.6 | 44.8 | 51.0 | 48.3 | 32.9 | 68.1 | 67.3 | 71.4 | 79.3 | 72.2 | 80.4 | 73.1 |
| DCC [24] | ResNet50 | 56.9 | 43.7 | 50.3 | 43.3 | 44.9 | 56.2 | 49.2 | 43.0 | 88.5 | 78.5 | 70.2 | 79.3 | 75.9 | 88.6 | 80.2 |
| OVANet [40] | | 56.0 | 47.1 | 51.7 | 44.9 | 47.4 | 57.2 | 50.7 | 53.1 | 85.8 | 79.4 | 80.1 | 95.4 | 84.0 | 94.3 | 86.5 |
| UMAD [26] | | 59.0 | 44.3 | 50.1 | 42.1 | 32.0 | 55.3 | 47.1 | 58.3 | 79.1 | 77.4 | 87.4 | 90.7 | **90.4** | **97.2** | 87.0 |
| GATE [8] | | 57.4 | 48.7 | 52.8 | 47.6 | 49.5 | 56.3 | 52.1 | 56.4 | 87.7 | 81.6 | 84.2 | 94.8 | 83.4 | 94.1 | 87.6 |
| UniOT [6] | | 59.3 | 47.8 | 51.8 | 46.8 | 48.3 | 58.2 | 52.0 | 57.3 | 83.7 | **85.3** | 71.4 | 91.2 | 70.9 | 90.84 | 82.2 |
| GLC [34] | | **63.3** | 50.5 | 54.9 | 50.9 | 49.6 | 61.3 | 55.1 | 73.1 | 81.5 | 84.5 | **89.8** | 90.4 | 88.4 | 92.3 | 87.8 |
| GLC [34] | ViT-B/16 | 51.2 | 44.5 | 55.6 | 43.1 | 47.0 | 39.1 | 46.8 | **80.3** | 80.5 | 80.4 | 77.5 | **95.6** | 77.7 | 96.9 | 84.8 |
| DCC [24] | | 61.1 | 38.8 | 51.8 | 49.3 | 49.1 | 60.3 | 52.2 | 61.2 | 82.2 | 76.9 | 83.6 | 75.2 | 85.8 | 88.7 | 82.1 |
| MemSPM+DCC | | 62.4 | **52.8** | **58.5** | **53.3** | **50.4** | **62.6** | **56.7** | 79.3 | 88.0 | 84.6 | 88.7 | 87.6 | 87.9 | 94.3 | **88.5** |

Table 3: H-score (%) comparison in UniDA scenario on Office-Home, some results are cited from [24, 34]

| Method | Backbone | Office-Home | | | | | | | | | | | | |
|---|---|---|---|---|---|---|---|---|---|---|---|---|---|---|
| | | Ar2Cl | Ar2Pr | Ar2Rw | Cl2Ar | Cl2Pr | Cl2Rw | Pr2Ar | Pr2Cl | Pr2Rw | Rw2Ar | Rw2Cl | Rw2Pr | Avg |
| UAN [47] | | 51.6 | 51.7 | 54.3 | 61.7 | 57.6 | 61.9 | 50.4 | 47.6 | 61.5 | 52.6 | 52.6 | 65.2 | 56.6 |
| CMU [14] | | 56.0 | 56.9 | 59.2 | 67.0 | 64.3 | 67.8 | 54.7 | 51.1 | 66.4 | 68.2 | 57.9 | 69.7 | 61.6 |
| DCC [24] | ResNet50 | 58.0 | 54.1 | 58.0 | 74.6 | 70.6 | 77.5 | 64.3 | 73.6 | 74.9 | 81.0 | 75.1 | 80.4 | 70.2 |
| OVANet [40] | | 62.8 | 75.6 | 78.6 | 70.7 | 68.8 | 75.0 | 71.3 | 58.6 | 80.5 | 76.1 | 64.1 | 78.9 | 71.8 |
| UMAD [26] | | 61.1 | 76.3 | 82.7 | 70.7 | 67.7 | 75.7 | 64.4 | 55.7 | 76.3 | 73.2 | 60.4 | 77.2 | 70.1 |
| GATE [8] | | 63.8 | 75.9 | 81.4 | 74.0 | 72.1 | 79.8 | 74.7 | 70.3 | 82.7 | 79.1 | 71.5 | 81.7 | 75.6 |
| UniOT [6] | | 67.2 | 80.5 | 86.0 | 73.5 | 77.3 | 84.3 | 75.5 | 63.3 | 86.0 | 77.8 | 65.4 | 81.9 | 76.6 |
| GLC [34] | | 64.3 | 78.2 | 89.8 | 63.1 | 81.7 | 89.1 | 77.6 | 54.2 | **88.9** | 80.7 | 54.2 | 85.9 | 75.6 |
| GLC [34] | ViT-B/16 | 68.5 | 89.8 | **91.0** | **82.4** | 88.1 | **89.4** | **82.1** | 69.7 | 88.2 | **82.4** | 70.9 | 88.9 | 82.6 |
| DCC [24] | | 62.6 | 88.7 | 87.4 | 63.3 | 68.5 | 79.3 | 67.9 | 63.8 | 82.4 | 70.7 | 69.8 | 87.5 | 74.4 |
| MemSPM+DCC | | **78.1** | **90.3** | 90.7 | 81.9 | **90.5** | 88.3 | 79.2 | **77.4** | 87.8 | 78.8 | **76.2** | **91.6** | **84.2** |

[46], a more difficult dataset consisting of 15500 images across 65 categories and 4 domains (Artistic images, Clip-Art images, Product images, and Real-World images); VisDA [33], a large-scale dataset with a synthetic source domain of 15K images and a real-world target domain of 5K images; and DomainNet [32], the largest domain adaptation dataset with approximately 600,000 images. Similar to previous studies [14], we evaluate our model on three subsets of DomainNet (Painting, Real, and Sketch).

As in previous work [24, 41, 2, 4, 47], we divide the label set into three groups: common classes $C$, source-private classes $\hat{C}_s$, and target-private classes $\hat{C}_t$. The separation of classes for each of the four datasets is shown in Table 1 and is determined according to alphabetical order.

Table 1: The division on label set, Common Class ($C$) / Source-Private Class ($\hat{C}_s$) / Target Private Class ($\hat{C}_t$).

| Dataset | Class Split($C/\hat{C}_s/\hat{C}_t$) | | |
|---|---|---|---|
| | PDA | OSDA | UniDA |
| Office-31 | 10 / 21 / 0 | 10 / 0 / 11 | 10 / 10 / 11 |
| OfficeHome | 25 / 40 / 0 | 25 / 0 / 40 | 10 / 5 / 50 |
| VisDA | 6 / 6 / 0 | 6 / 0 / 6 | 6 / 3 / 3 |
| DomainNet | —— | —— | 150 / 50 / 145 |

**Evaluation Metrics**. We report the averaged results of three runs. For the PDA scenario, we calculate the classification accuracy over all target samples. The usual metrics adopted to evaluate OSDA are the average class accuracy over the known classes $OS^*$, and the accuracy of the unknown class $UNK$. In the OSDA and UniDA scenarios, we consider the balance between "known" and "unknown" categories and report the H-score [1]:

$$\text{H-score} = 2 \times \frac{OS^* \times UNK}{OS^* + UNK}, \tag{9}$$

which is the harmonic mean of the accuracy of "known" and "unknown" samples.

**Implementation Details**. Our implementation is based on PyTorch [31]. We use ViT-B/16 [12] as the backbone pretrained by CLIP [35] for the MemSPM is hard to train with a randomly initialized encoder. The classifier consists of two fully-connected layers, which follows the previous design [4, 47, 41, 14, 24]. The weights in the $\mathcal{L}$ are empirically set as $\lambda_1 = 0.1$, $\lambda_2 = 3$ and $\lambda_3 = 0.5$ fellow DCC [24]. For a fair comparison, we also adopt ViT-B/16 as backbone for DCC [24] and state-of-art method GLC [34]. We use the official code of DCC [24] (https://github.com/Solacex/Domain-Consensus-Clustering) and GLC [34] (https://github.com/ispc-lab/GLC).

Table 4: H-score (%) comparison in OSDA scenario on Office-Home, VisDA and Office-31, some results are cited from [24, 34]

| Method | Backbone | Office-Home | | | | | | | | | | | | | Office-31 | VisDA |
|---|---|---|---|---|---|---|---|---|---|---|---|---|---|---|---|---|
| | | Ar2Cl | Ar2Pr | Ar2Rw | Cl2Ar | Cl2Pr | Cl2Rw | Pr2Ar | Pr2Cl | Pr2Rw | Rw2Ar | Rw2Cl | Rw2Pr | Avg | Avg | Avg |
| OSBP [41] | ResNet50 | 55.1 | 65.2 | 72.9 | 64.3 | 64.7 | 70.6 | 63.2 | 53.2 | 73.9 | 66.7 | 54.5 | 72.3 | 64.7 | 83.7 | 52.3 |
| CMU [14] | | 55.0 | 57.0 | 59.0 | 59.3 | 58.2 | 60.6 | 59.2 | 51.3 | 61.2 | 61.9 | 53.5 | 55.3 | 57.6 | 65.2 | 54.2 |
| DCC [24] | | 56.1 | 67.5 | 66.7 | 49.6 | 66.5 | 64.0 | 55.8 | 53.0 | 70.5 | 61.6 | 57.2 | 71.9 | 61.7 | 72.7 | 59.6 |
| OVANet [40] | | 58.6 | 66.3 | 69.9 | 62.0 | 65.2 | 68.6 | 59.8 | 53.4 | 69.3 | 68.7 | 59.6 | 66.7 | 64.0 | 91.7 | 66.1 |
| UMAD [26] | | 59.2 | 71.8 | 76.6 | 63.5 | 69.0 | 71.9 | 62.5 | 54.6 | 72.8 | 66.5 | 57.9 | 70.7 | 66.4 | 89.8 | 66.8 |
| GATE [8] | | 63.8 | 70.5 | 75.8 | 66.4 | 67.9 | 71.7 | 67.3 | 61.5 | 76.0 | 70.4 | 61.8 | 75.1 | 69.0 | 89.5 | 70.8 |
| ROS [6] | | 60.1 | 69.3 | 76.5 | 58.9 | 65.2 | 68.6 | 60.6 | 56.3 | 74.4 | 68.8 | 60.4 | 75.7 | 66.2 | 85.9 | 66.5 |
| GLC [34] | | 65.3 | 74.2 | 79.0 | 60.4 | 71.6 | 74.7 | 63.7 | 63.2 | 75.8 | 67.1 | 64.3 | 77.8 | 69.8 | 89.0 | 72.5 |
| GLC [34] | ViT-B/16 | 68.4 | 81.7 | 84.5 | **76.0** | 82.4 | **83.8** | 69.9 | 59.6 | 84.6 | **73.3** | **66.8** | 83.9 | 76.2 | 90.1 | **81.6** |
| DCC [24] | | 62.9 | 73.3 | 78.4 | 49.8 | 69.2 | 75.0 | 59.3 | 61.5 | 80.9 | 68.1 | 62.5 | 80.0 | 68.4 | 81.9 | 66.2 |
| MemSPM+DCC | | **69.7** | **83.2** | **85.2** | 72.0 | 79.2 | 81.2 | **72.3** | **66.7** | **85.2** | 72.7 | 66.0 | **84.5** | **76.5** | **95.6** | 79.7 |

Table 5: H-score (%) comparison in PDA scenario on Office-Home, VisDA and Office-31, some results are cited from [24, 34]

| Method | Backbone | Office-Home | | | | | | | | | | | | | Office-31 | VisDA |
|---|---|---|---|---|---|---|---|---|---|---|---|---|---|---|---|---|
| | | Ar2Cl | Ar2Pr | Ar2Rw | Cl2Ar | Cl2Pr | Cl2Rw | Pr2Ar | Pr2Cl | Pr2Rw | Rw2Ar | Rw2Cl | Rw2Pr | Avg | Avg | Avg |
| ETN [5] | ResNet50 | 59.2 | 77.0 | 79.5 | 62.9 | 65.7 | 75.0 | 68.3 | 55.4 | 84.4 | 75.7 | 57.7 | 84.5 | 70.4 | 96.7 | 59.8 |
| BA3US [27] | | 60.6 | **83.2** | **88.4** | 71.8 | 72.8 | 83.4 | 75.5 | 61.6 | 86.5 | 79.3 | 62.8 | **86.1** | **76.0** | **97.8** | 54.9 |
| DCC [24] | | 54.2 | 47.5 | 57.5 | **83.8** | 71.6 | **86.2** | 63.7 | **65.0** | 75.2 | **85.5** | **78.2** | 82.6 | 70.9 | 93.3 | 72.4 |
| OVANet [40] | | 34.1 | 54.6 | 72.1 | 42.4 | 47.3 | 55.9 | 38.2 | 26.2 | 61.7 | 56.7 | 35.8 | 68.9 | 49.5 | 74.6 | 34.3 |
| UMAD [26] | | 51.2 | 66.5 | 79.2 | 63.1 | 62.9 | 68.2 | 63.3 | 56.4 | 75.9 | 74.5 | 55.9 | 78.3 | 66.3 | 89.5 | 68.5 |
| GATE [8] | | 55.8 | 75.9 | 85.3 | 73.6 | 70.2 | 83.0 | 72.1 | 59.5 | **84.7** | 79.6 | 63.9 | 83.8 | 74.0 | 93.7 | 75.6 |
| GLC [34] | | 55.9 | 79.0 | 87.5 | 72.5 | 71.8 | 82.7 | **74.9** | 41.7 | 82.4 | 77.3 | 60.4 | 84.3 | 72.5 | 94.1 | 76.2 |
| GLC [34] | ViT-B/16 | 63.2 | 80.7 | 86.5 | 76.0 | 77.9 | 84.1 | 74.5 | 56.8 | **84.7** | 79.8 | 57.4 | 83.0 | 75.4 | 91.5 | 86.2 |
| DCC [24] | | 59.4 | 78.8 | 83.2 | 61.95 | **78.6** | 79.3 | 64.2 | 44.4 | 82.9 | 76.5 | 70.7 | 84.6 | 72.1 | 93.7 | 79.8 |
| MemSPM+DCC | | **64.7** | 81.1 | 84.5 | 74.8 | 74.7 | 77.5 | 58.7 | 60.3 | 84.2 | 70.3 | 77.2 | 85.8 | 74.5 | 94.4 | **87.9** |

## 4.2 Comparison with State-of-The-Arts

We compare our method with previous state-of-the-art algorithms in three sub-cases of unsupervised domain adaptation, namely, object-specific domain adaptation (OSDA), partial domain adaptation (PDA), and universal domain adaptation (UniDA). In UniDA, we compare our method to previous universal domain adaptation approaches, which do not take into account the prior that private classes exist only in either the source domain (PDA) or the target domain (OSDA). Additionally, we compare our method to the OSDA and PDA baselines that consider the prior information unique to each sub-case.

**Results on UniDA**. In the most challenging setting, i.e. UniDA, our MemSPM approach achieves the state-of-the-art performance. Table 2 shows the results on DomainNet, VisDA and Office-31, and result of Office-Home is summarized in Table 3. We mainly compare with GLC and DCC using ViT-B/16 as backbone. On Office-31, the MemSPM+DCC outperform previous state-of-art method GLC by 3.7% and surpasses the DCC by 6.4%. On visda, our method surpasses the DCC by a huge margin of 16.1%. Our method also surpasses the GLC by 9.9% and the DCC by 4.5% on DomainNet. On the Office-Home, we surpasses the DCC by 9.8% and the GLC by 3.7%.

**Results on OSDA and PDA**. In table 4 and table 5, we present the results on Office-Home, Office-31 and VisDA under OSDA and PDA scenarios. In the OSDA scenario, MemSPM+DCC still achieves state-of-the-art performance. Specifically, MemSPM+DCC obtains 95.6% H-score on Office-31, with an improvement of 5.5% compared to GLC and 13.7% compared to DCC. In the PDA scenario, MemSPM still achieves comparable performance compared to methods tailored for PDA. The MemSPM+DCC surpasses the DCC by 8.1% on the VisDA.

## 4.3 Ablation Studies

**Visualization with Reconstruction and tSNE** We first visualize what the memory learns from Office-Home by sampling a single sub-prototype and adapting an auxiliary reconstruction task: $X \rightarrow \hat{X}$. We also provide the tSNE of the $\hat{Z}$ which retrieving the most related sub-prototypes. The visualization is shown in Figure 3. The tSNE visualization depicts the distribution of sub-classes within each category, indicative of MemSPM's successful mining of sub-prototypes. The reconstruction visualization shows what have been learned by MemSPM, demonstrating its ability to capture intra-class diversity.

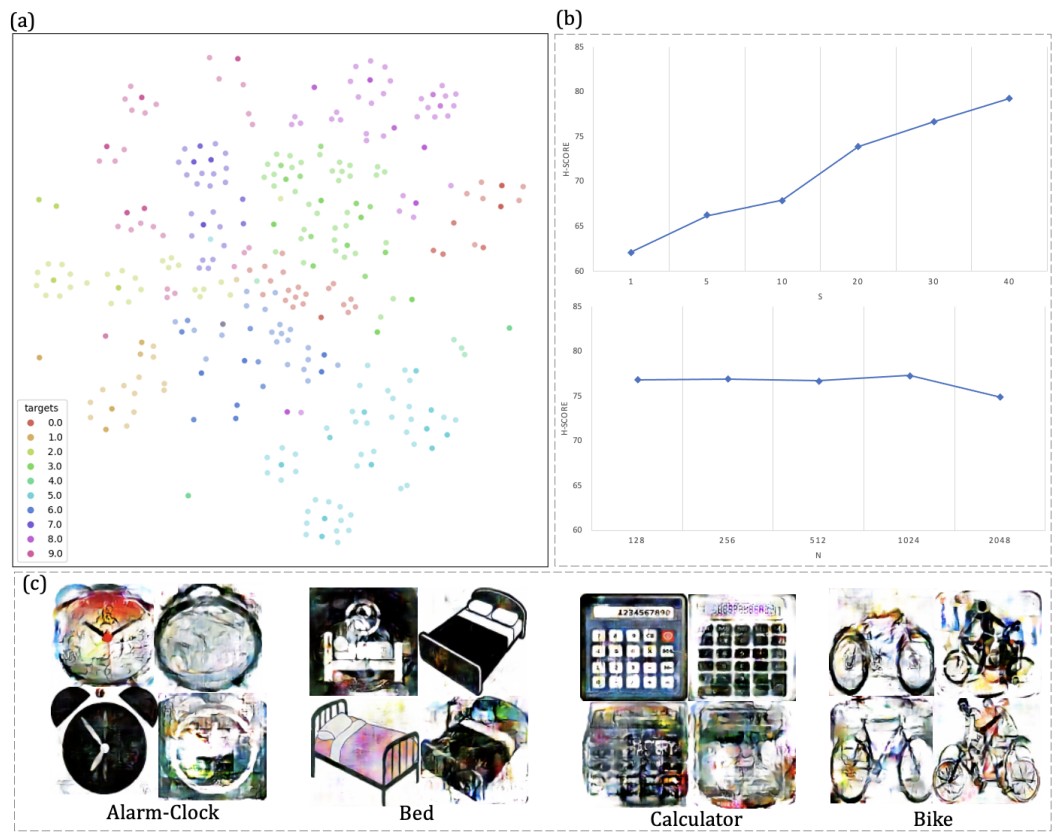

Figure 3: (a) The tSNE visualization shows the feature space of the sub-classes belonging to the each category, which demonstrate the MemSPM mining the sub-prototypes successfully. (b) The results of different values of $S$ and $N$. (c) The reconstruction visualization shows what have been learned in the memory, which demonstrate the intra-class diversity have been learned by MemSPM.

**Effect of Memory-Assisted Sub-Prototype Mining**. As the results shown in table 2, table 3, table 4 and table 5, the MemSPM+DCC evaluted on four benchmarks has surpassed the DCC on UniDA, OSDA and PDA scenarios. The MemSPM can significantly improve the performance of the DCC when using ViT-B/16 as backbone. The reason for utilizing the ViT-B/16 is that the memory module of the MemSPM with huge latent space is initialized by randomly normal distribution, which make it hard to retrieve the different sub-prototypes at early stages of training. So, we need ViT as backbone, which have learned a more global feature space.

**Sensitivity to Hyper-parameters**. We conducted experiments on the VisDA dataset under the UniDA setting to demonstrate the impact of hyperparameters $S$ and $N$ on the performance of our method. The impact of $S$ are shown in Figure 3. When $S \geq 20$, the performance achieve a comparable level. At the same time, the performance of the model is not sensitive to the value of $N$, when $S = 30$.

# 5   Conclusion

In this paper, we propose the Memory-Assisted Sub-Prototype Mining (MemSPM) method, which can learn the intra-class diversity by mining the sub-prototypes to represent the sub-classes. Compared with the previous methods, which overlook the intra-class structure by using one-hot label, our Mem-SPM can learn the class feature from a more subdivided sub-class perspective to improve adaptation performance. At the same time, the visualization of the tSNE and reconstruction demonstrates the sub-prototypes have been well learned as we expected. Our MemSPM method exhibits superior performance in most cases compared with previous state-of-the-art methods on four benchmarks.

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
