# Appendix

In the supplementary material, we provide additional visualization results, limitations, potential negative societal impacts and compute requirements of the MemSPM. In the pursuit of reproducible research, we will make the demo and network weights of our code available to the public.

This supplementary is organized as follows:

## A    Notations

Table 1:

| | Symbol | Description |
|---|---|---|
| Model | $f_{encode}^{fixed}(\cdot)$ | Fixed image encoder |
| | $f_{decode}^{unfixed}(\cdot)$ | Unfixed reconstruction decoder |
| | $f_{class}^{UniDA}$ | UniDA classifier |
| | $M$ | Memory unit |
| | $W$ | Weight vector |
| Space | $\mathcal{D}^s$ | Labeled source dataset |
| | $\mathcal{D}^t$ | Unlabeled target dataset |
| | $C$ | Common label set |
| | $C_s$ | Source label set |
| | $C_t$ | Target label set |
| | $\hat{C}_s$ | Source private label set |
| | $\hat{C}_t$ | Target private label set |
| Samples | $X$ | Input image |
| | $\hat{X}$ | Reconstruction of image |
| | $Z$ | Input-oriented embedding |
| | $\hat{Z}$ | Task-oriented embedding |
| | $L$ | Label of the image |
| | $\hat{L}$ | Prediction of image |
| Measures | $w_{i,j}$ | Attention weight measurement between $Z$ and sub-prototype |
| | $d(\cdot,\cdot)$ | Cosine similarity measurement |
| | $\hat{w_{i,j}}$ | Adaptive threshold operation on $w_{i,j}$ |
| Hyperparameters | $N$ | Number of memory items |
| | $S$ | Number of sub-prototypes partitioned in each memory item |
| | $D$ | Dimension of each sub-prototype |
| | $K$ | Top-K relevant sub-prototypes of $Z$ |

## B    Limitation

Training memory unit of MemSPM is challenging when adopting the commonly used ResNet-50 as the backbone. This is due to the memory unit's composition of massive randomly initialized tensors.

During the early stage of training, there is a lack of discriminability in the input-oriented embedding, which leads to addressing only a few sub-prototypes. This decoupling of the memory unit from the input data necessitates using a better pre-trained model (ViT-B/16 pre-trained on CLIP) and fixing the encoder to reduce computation requirements. Additionally, the number of sub-prototypes in one memory item might need to be adjusted for the diversity of the category.

## C   Potential Societal Impact

Our finding of the intra-class concept shift may influence the future work on domain adaption or other tasks. They can optimize the construction and refinement of the feature space by considering the intra-class distinction. The MemSPM also provides a method can be used to demonstrate the interpretability of model for further deployment. However, the utilization of MemSPM method for illegal purposes may be facilitated by their increased availability to organizations or individuals. And the MemSPM method may be susceptible to adversarial attacks as all contemporary deep learning systems. Although we demonstrate increased performance and interpretability compared to the state-of-the-art methods, negative transfer is still possible in extreme cases of domain-shift or category-shift. Therefore, our technique should not be employed in critical applications or to make significant decisions without human supervision.

## D   Implementation details

**DCC.** We use ViT-B/16 [1] as the backbone. The classifier is made up of two FC layers. We use Nesterov momentum SGD to optimize the model, which has a momentum of $0.9$ and a weight decay of 5e-4. The learning rate decreases by a factor of $(1 + \alpha \frac{i}{N})^{-\beta}$ , where $i$ and $N$ represent current and global iteration, respectively, and we set $\alpha = 10$ and $\beta = 0.75$. We use a batch size of 36 and the initial learning rate is set as 1e-4 for Office-31, and 1e-3 for Office-Home and DomainNet. We use the settings detailed in [2]. PyTorch [3] is used for implementation.

**GLC.** We use ViT-B/16 [1] as the backbone. The SGD optimizer with a momentum of $0.9$ is used during the target model adaptation phase of GLC [6]. The initial learning rate is set to 1e-3 for Office-Home and 1e-4 for both VisDA and DomainNet. The hyperparameter $\rho$ is fixed at $0.75$ and $|L|$ at 4 across all datasets, while $\eta$ is set to $0.3$ for VisDA and $1.5$ for Office-Home and DomainNet, which corresponds to the settings detailed in [6]. PyTorch [3] is used for implementation.

**Existing code used.**

- DCC [2]: `https://github.com/Solacex/Domain-Consensus-Clustering`
- GLC [6]: `https://github.com/ispc-lab/GLC`
- PyTorch [3]: `https://pytorch.org/`

**Existing datasets used.**

- Office-31 [7]: `https://www.cc.gatech.edu/âĹijjudy/domainadapt`
- Office-Home [8]: `https://www.hemanthdv.org/officeHomeDataset.html`
- DomainNet [4]: `http://ai.bu.edu/M3SDA`
- VisDA [5]: `http://ai.bu.edu/visda-2017/`

**Compute Requirements.** For our experiments, we used a local desktop machine with an Intel Core i5-12490f, a single Nvidia RTX-3090 GPU and 32GB of RAM. When we adapt the batch-size used in DCC [2], our MemSPM only occupies 4GB of GPU memory during training in result of fixing the encoder.

## E   Visualization

We provid more results of visualization in Figure 1 and Figure 2 to reveal sub-prototypes stored in the memory unit, which demonstrate that our MemSPM approach can learn the intra-class concept shift.

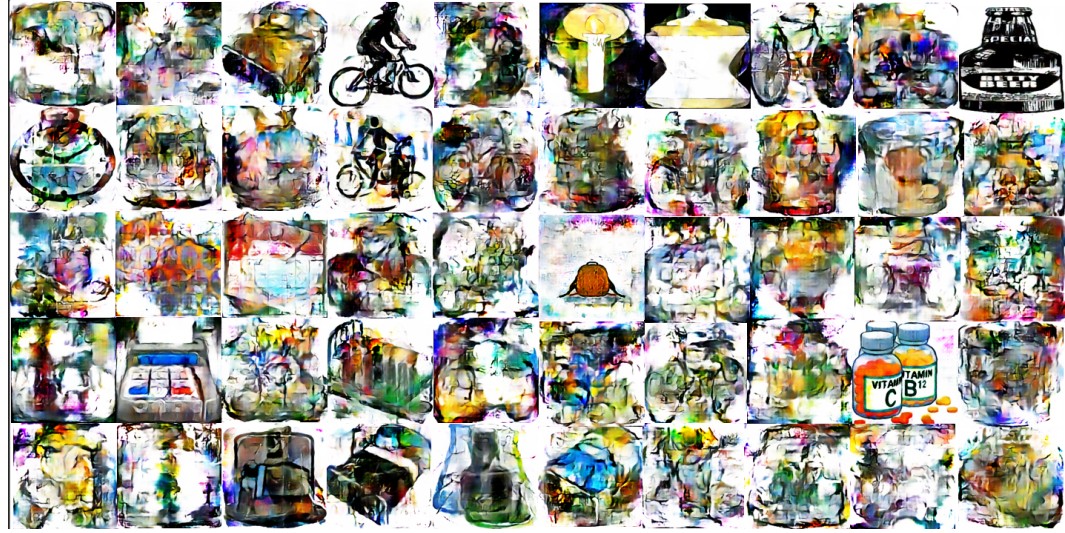

Figure 1: The reconstruction visualization shows what have been learned in the memory, which demonstrates the intra-class diversity have been learned by MemSPM.

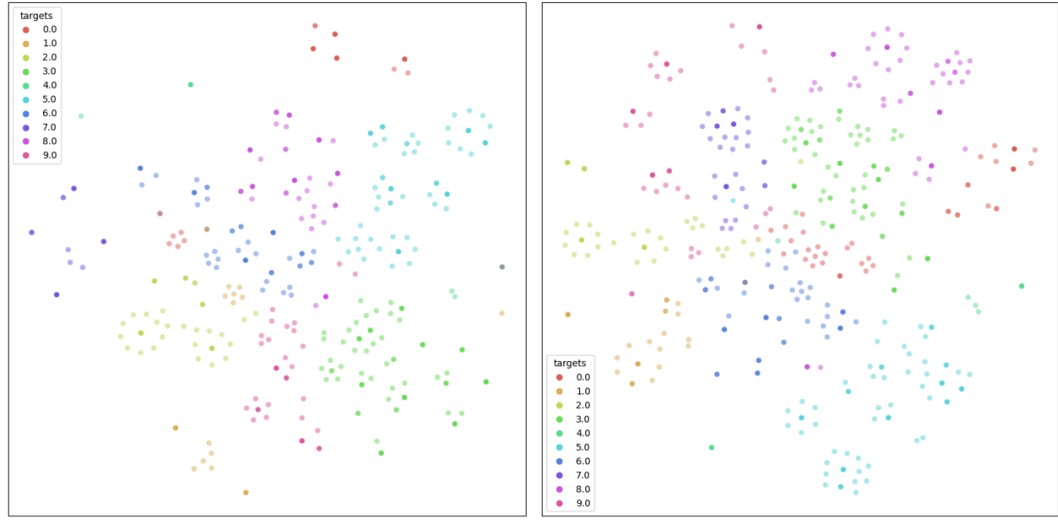

Figure 2: The tSNE visualization shows the distribution of the retrieved sub-prototypes and demonstrates that the sub-classes have been learned by MemSPM.