# OpenReview forum: "Memory-Assisted Sub-Prototype Mining for Universal Domain Adaptation"
_NeurIPS.cc/2023/Conference — Submitted to NeurIPS 2023_

### Official Review · Reviewer_QTwQ · 2023-06-12

**Soundness:** 1 poor
**Presentation:** 2 fair
**Contribution:** 2 fair
**Rating:** 3
**Confidence:** 5

**Summary:**

This paper aims to improve previous Universal Domain Adptation (UniDA) methods by further exploting the intra-class discrimination. For that, they propose a Memory-Assisted Sub-Prototype Mining (MemSPM) method. MemSPM learns to retrieve new task-oriented features given the input embedding features, and apply existing UniDA methods to the retrieving features. The paper also proposes an additional reconstruction task for the demonstration to the explainability of its proposed method as the authors claimed. Experiments on four datasets are conducted on three DA settings.

**Strengths:**

Considering the effect of learning intra-class discrimination for UniDA is indeed an interesting idea to focus on, and such motivation is new in the UniDA community. By exploiting the intra-class structure, the proposed MenSPM is somehow novel to see.

**Weaknesses:**

Although the motivation from exploiting intra-class structure is interesting to UniDA, the analysis and the evidences to support the effectiveness of such idea is not enough. This is mainly due to the following concerns.

1. Subclasses learning brings additional learning challenge and increases the learning cost to the problem, and not always the case that some classes have obvious subclasses, thus it is hard to say whether forcing subclasses learning would be beneficial to UniDA. To investivage this, I think it should have a solid analysis to the problem.

2. The proposed method introduces too many hyper-parameters to the leanning process, inlcuding $N$, $S$, $K$, $\lambda$, $\lambda_1$, $\lambda_2$, and $\lambda_3$, etc., and there have not sufficient studies to investigate those hyper-parameters for different datasets or tasks. Note that this is important in UniDA since there is no validation set for model selection. Therefore, it is hard to say whether the effectiveness of the method may come from hyper-parameters tunning.

3. Abalation studies are also not enough to understanding the effectiveness of different loss terms in Equation (8). Although improvements have shown when comparing to the DCC method, but to my knowledge with the CLIP models,  a simple baseline of standard training on source data only may already outperform the proposed method. However, this is not compared in the experiments.

4. The results reported in the ResNet50 are meaningless since the proposed method do not run on this backbone. This is also a limitation of the proposed method.

5. The experiments to verify the effectiveness of the proposed idea only conduct on the DCC method, which is not enough.

The authors claim that the proposed method could make interpretability from Figure 3, but I do not know how it works for the explainability since reconstruction does not imply interpretability. A random noise could also reconstruct the input.

The loss of $\mathcal{L}_{cdd}$ is not illustrated in the paper. It is a bad way to let readers to understand it from other papers as it is not popular.

Some typos exist in the paper, and please carefully check if some formulas are presented correctly, e.g., Equations (2), (6).

**Questions:**

All weaknesses listed above should be well addressed to improve the paper.

**Limitations:**

The authors have shown some limitations of the proposd method, but more should consider other that the method itself.

---

> ### Author Rebuttal · Authors · 2023-08-09
>
> ##### Re_Q1:
>
> Although the MemSPM has some additional learning challenges and costs, it is worth the cost because 4 commonly used datasets have significant concept shift in 90% of categories (Samples shown in **Figure 1(a)** ), and other larger dataset such as ImageNet even has more intra-class distinction. The MemSPM can learn the intra-class distinction and mine "sub-prototypes" for better alignment and adaptation. Moreover, our MemSPM will not force the sample of one category to a certain number of sub-classes. It is an adaptive process; the sample will have an unused memory item to represent when the sample has not paired sub-prototypes embedding from existing sub-classes.
>
> ***
>
> ##### Re_Q2:
>
> We conducted ablation experiments on hyperparameters, the results of which can be seen in **Figure 3(b)**. We found that MemSPM was insensitive to N, while larger S values resulted in better performance. For these experiments, we discovered that we only need to set a large S and N, and this setting depends solely on the GPU memory.
>
> We have conducted experiments on the contribution of each loss. The L_ce is for the classification that can not be removed. The L_cdd is removed and the model achieves **79.8%** which is **4.4%** lower than using the full loss function. After the experiment, we found that the coefficient of L_ce **($\lambda_{1} = 0.1$)** and the coefficient of L_cdd **($\lambda_{2} = 3$)**, which were provided by DCC had the best performance.
>
> For reconstruction task loss (L_rec), it wasn’t a main part of our loss. And in our experiment, we find that L_rec makes a very small impact on the performance, which is just for visualization. As we are mainly concerned with hyperparameters of memory structure, these ablation results of loss had no space to list in. For hyperparameters of the loss function, we fixed them for all datasets. Therefore, setting the hyperparameters does not require hyperparameters turning and any information about the target domain. Thanks for the advice, we will add more details in the revised manuscript.
>
> | Method | Backbone Pretrain | Ar2Cl | Ar2Pr | Ar2Rw | Cl2Ar | Cl2Pr | Cl2Rw | Pr2Ar | Pr2Cl | Pr2Rw | Rw2Ar | Rw2Cl | Rw2Pr | Avg  |
> |:-------:|:-------:|:-------:|:-------:|:-------:|:-------:|:-------:|:-------:|:-------:|:-------:|:-------:|:-------:|:-------:|:-------:|:------:|
> | CLIP-Baseline |ViT-B/16 CLIP|  64.6 |   84.3    |    78.1   |   73.7    |   88.2    |    86.5   |    68.1   |   68.7    |  **89.6**     |   68.5    |   69.4    |  86.6    | 77.2 |
> | DCC+MemSPM Without Lcdd |ViT-B/16 CLIP|  75.9 |   75.4    |    86.4   |   80.1    |   71.6    |    87.5   |    70.1   |   **87.1**    |  88.7     |   74.2    |   73.5    |   88.8   | 79.8|
> | DCC+MemSPM |ViT-B/16 CLIP |   **78.1**    |   **90.3**    |   **90.7**    |   **81.9**   |    **90.5**   |   **88.3**    |   79.2    |   77.4    |  87.8     |   **78.8**   |  **76.2**     |  **91.6**   | **84.2** |
>
>
> ***
>
> ##### Re_Q3:
>
> We have conducted experiments on the contribution of each loss. The L_ce is for the classification that cannot be removed. The L_cdd is removed and the model achieves **79.8%** which is **4.4%** lower than using the full loss function.
>
> CLIP-based embedding does have some cross domains knowledge but it was still influenced by the larger domain gap. We have tested the simple baseline of CLIP on officehome reaching **77.2 %** (H-score), which still has **4.4%** lower than our MemSPM.
>
> In **Table 2** and **Table 3**, the results of GLC, DCC, and MemSPM in the bottom all used the ViT-B/16 (pre-trained by the CLIP model), so the comparison is fair. The GLC was the SOTA result of UniDA and DCC is part of our code base. When these two methods are applied to CLIP encoder, they must perform better than a simple CLIP baseline.
>
> ***
>
> ##### Re_Q4:
>
> Since these results of typical methods were presented in many related works, so we also keep them in the table for a complete comparison. We conducted experiments on GLC and DCC with ViT-B/16 (pre-trained by the CLIP model) for comparison. The GLC was the SOTA method in UniDA and the DCC was part of our code base, so the comparison is fair and effective.
>
> ***
>
> ##### Re_Q5:
>
> The MemSPM indeed can be used in other methods, but we choose the DCC that most fits our motivation in the UniDA task. The experiments listed in **Tables 2,3,4, and 5** have proved the effectiveness of our MemSPM. Thanks for the advice. We will apply the MemSPM to other methods in the revision.
>
> ***
>
> ##### Re_Q6:
>
> The embeddings used to reconstruct images are from memory items. The memory items are learned by comparing them with the input-oriented embedding, so they are very different from random noise. As shown in **Figure 3**, the t-SNE visualization shows that the sub-prototypes from memory have learned the sub-classes knowledge. So, the reconstruction of these sub-prototypes is used to show that sub-prototypes have learned the representative features of sub-classes.
>
> ***
>
> ##### Re_Q7:
>
> Thanks for the advice. We will add more details of the L_cdd in the revised manuscript.
>
> ***
>
> ##### Re_Q8:
>
> Thanks for the advice. We have carefully fixed these typos.

---

### Official Review · Reviewer_pJBT · 2023-06-27

**Soundness:** 3 good
**Presentation:** 2 fair
**Contribution:** 2 fair
**Rating:** 6
**Confidence:** 5

**Summary:**

This work proposes to exploit the intrinsic structures for each class, where sub-prototypes are devised to associate domain-common knowledge for universal domain adaptation. Specifically, MemSPM employs a memory module to mine sub-class information, and a corresponding reconstruction module to derive task-oriented representations. Experiments on representative benchmarks are conducted to verify the effectiveness of the proposed approach.

**Strengths:**

1, This paper is generally well-written and easy to follow, and neat figures are presented to enable a more intuitive understanding.

2, The motivation for decoupling with subclass structures seems reasonable.

3, The technical details are well explained.

4, Surpassing previous methods with noticeable margins, justifying its effectiveness.

**Weaknesses:**

I think the main drawback of this paper lies in its presentations:

1, Motivations of some designs are not well explained, i.e., why sub-prototypes benefits the universal scenario？

2, Some technical details seem missing.

The details of these concerns are presented in the ‘Questions’ part.

Minors:
Page 5 Line 179: missing space ''[17]that''


**Questions:**

1, Why can sub-prototypes benefit the universal domain adaptation scenario?
I understand that, even within a domain, samples from the same class can be grouped into sub-classes. But, a critical part is missing why this helps the cross-domain association of common classes. which is the core problem for universal domain adaptation. An explanation or empirical justification is needed here, i.e., what is the pattern of retrieved sub-prototypes for common samples and private ones?

2, Some technical details are not comprehensive enough.
1) Is the memory learnable parameters? How to initialize them? This can be basic knowledge for people familiar with this, but it is still necessary to briefly detail this.
2) After reading sec 3.5,  it is still unclear to be how the sub-prototypes help align the embeddings \hat{Z}.

3, In Fig. 1 (c), does this method assume the sub-class of two domains can be matched? This seems unrealistic under the distribution shift.



**Limitations:**

Yes.

---

> ### Author Rebuttal · Authors · 2023-08-08
>
> ##### Q1:
>
> Why can sub-prototypes benefit the universal domain adaptation scenario? I understand that, even within a domain, samples from the same class can be grouped into sub-classes. But, a critical part is missing why this helps the cross-domain association of common classes. which is the core problem for universal domain adaptation. An explanation or empirical justification is needed here, i.e., what is the pattern of retrieved sub-prototypes for common samples and private ones?
>
> ##### Re:
>
> As our motivation illustrated in **Figure 1**, the samples that are annotated as one category usually have significant intra-class differences. However, in previous work, they just forced them to align together for adaptation. So, these methods are more likely to classify unknown classes into known classes incorrectly. In the feature space, samples’ features of different sub-classes still have gaps in the feature space, so it is not reasonable to align samples from different sub-classes together not only in human understanding but also in the learned feature space.
>
> For these reasons, if we can have the target domain sample aligned in the sub-class level with the source domain sample, we can avoid the drawback of aligning two samples that are very different together and make the adaption more reasonable.
>
> The pattern of retrieved sub-prototypes is based on comparing the similarity of input-oriented embedding with the memory items. Given a sample of common classes, it will find some similar sub-prototypes from the learned memory to create task-oriented embedding for the downstream classification task. However, for the case of private classes, it will get some redundant memory items that have not been used by the source domain, thus their task-oriented embedding will be far from common classes’ embedding.
>
> ***
>
> ##### Q2:
>
> Some technical details are not comprehensive enough.
>
> 1. Is the memory learnable parameters? How to initialize them? This can be basic knowledge for people familiar with this, but it is still necessary to briefly detail this.
> 2. After reading sec 3.5, it is still unclear to be how the sub-prototypes help align the embeddings \hat{Z}.
>
> ##### Re:
>
> 1. Thanks for the advice. The memory structure has learnable parameters and we only use the uniform distribution to initialize memory items. We will add these to the revised manuscript.
> 2.  In our approach, the memory learns sub-prototypes that embody sub-classes learned from the source domain. During testing of the target samples, the encoder produces embedding that is compared to source domain sub-prototypes learned in the memory. Subsequently, an embedding for the query sample is generated through weighted sub-prototype sampling in the memory. This results in reduced domain shifts before the embedding give into the classifier. The Cycle-Consistent Alignment and Adaption is a method that matches the target domain sub-prototype clusters to the source domain and then aligns two similar sub-classes together. Due to space constraints, we do not describe the details presented in the DCC, we will make the clarification clearer in the revised manuscript.
>
> ***
>
> ##### Q3:
>
> In Fig. 1 (c), does this method assume the sub-class of two domains can be matched? This seems unrealistic under the distribution shift.
>
> ##### Re:
>
> In the visualization results of **Figure 3(a)**, we can find that the MemSPM has learned the knowledge of sub-classes.  When the MemSPM comes into a new domain, the samples of the target domain can match with the sub-prototypes of each sub-classes. After that, the sampled sub-prototypes of the source domain will be used to represent the target domain input. Therefore, we can reduce the distribution shift between the task-oriented embedding of two domains.
>
> To demonstrate this assumption, we tested visually similar samples from two domains and find that the memory addressing module can mostly sample the same sub-prototypes from the memory. This experiment demonstrates that the sub-classes of two domains can be matched. We will add this part of the analysis to the revised manuscript.

---

> > ### Comment · Reviewer_pJBT · 2023-08-18
> > **Response to the rebuttal**
> >
> > Thanks for the rebuttal.
> > The answers convince me and clarify some technical details.
> > Based on the rebuttal, I decide to raise my score.

---

> > > ### Author Response · Authors · 2023-08-18
> > >
> > > Thank you for your feedback and efforts on our work.

---

### Official Review · Reviewer_EAMn · 2023-07-04

**Soundness:** 3 good
**Presentation:** 2 fair
**Contribution:** 3 good
**Rating:** 6
**Confidence:** 5

**Summary:**

This paper focuses on Universal Domain Adaptation (UniDA), a practical DA setting that does not make any assumptions on the relation between source and target label sets. The goal is to adapt a classifier from source to target domain such that both source and target domains may have their own private classes apart from shared classes. The paper claims that existing UniDA methods overlook the intrinsic structure in the categories, which leads to suboptimal feature learning and adaptation. Hence, they propose memory-assisted sub-prototype mining (MemSPM) that learns sub-prototypes in a memory mechanism to embody the subclasses from the source data. Then, for target samples, weighted sub-prototype sampling is used before passing the embedding to a classifier, which results in reduced domain shift for the embedding. They also propose an adaptive thresholding technique to select relevant sub-prototypes. Finally, they adopt the cycle consistent matching loss objective from DCC [24] along with an auxiliary reconstruction loss for training. They show results on UniDA, Partial DA, and Open-Set DA using standard benchmarks like Office-31, Office-Home, VisDA, and DomainNet.

**Strengths:**

* The motivating ideas for the approach are interesting and intuitive. Further, the technical contributions are novel as well as effective.

* It is intriguing that the auxiliary reconstruction task provides interpretability, which is usually not possible in existing DA solutions.

* The paper is fairly easy to follow (with the exception of some equations and many typos and grammatical errors, see Weaknesses).

* With their method and the advantages of a CLIP-pretrained ViT model, they achieve large improvements over existing ResNet-based methods. While they also show small improvements over some existing methods using the CLIP-pretrained model, this can serve as a new strong baseline for future UniDA work.

**Weaknesses:**

* The paper claims that existing UniDA works overlook the internal intrinsic structure in the categories.
    * However, [W1] aims to resolve the same problem. [W1] proposes to learn lower-level visual primitives that are unaffected by the category shift in the higher-level features. And, in their proposed word-prototype-space, different visual primitives can be shared across domains and classes (including unknown classes).
    * There is a significant overlap in the motivation given by this paper and that of [W1]. Consequently, the high-level conceptual novelty of this paper is overclaimed. However, I do believe that these conceptual ideas are interesting as well as important for UniDA.
    * Please discuss the similarities and differences (both in terms of motivation and the actual approach) of this paper w.r.t. [W1].
    * Another paper with similar conceptual ideas is [W2].

* This paper lacks some mathematical rigor.
    * Eq. 1, 2: $\hat{Z}=W\cdot M$ is shown as matrix multiplication (I assume that it is not element-wise multiplication since dimensions of $W$ and $M$ are different), but the expansion of this matrix multiplication contains an arg-max over the elements of $W$. Then, it does not make sense for the overall computation to be a standard matrix multiplication.
    * Eq. 1, 2: the text mentions that $s_i$ is the index of sub-prototypes in the $i^\text{th}$ item but Eq. 2 implies that $s_i$ is a particular dimension found with arg-max. This seems contradictory and is confusing.
    * Eq. 2: Use $\mathop{\arg\max}_{j}$ instead of using `dim=1` since it is a mathematical equation and not the code implementation.
    * Eq. 5: It is unclear which dimension is used for top-$k$
    * Eq. 6: It should be $\max(... , 0)$ instead of just $\max(...)$.

* The requirement of a CLIP-pretrained backbone is very restrictive since the method cannot be extended to other settings (like medical imaging) where the CLIP-pretraining may be suboptimal. While the paper shows comparisons where prior methods use the CLIP-pretrained model, it should also show comparisons when starting from a random initialization as well as the more widely used ImageNet initialization.
    * The paper claims that a CLIP backbone is needed to retrieve sub-prototypes in early iterations. Why not start retrieving sub-prototypes after a few epochs of normal training?

* L135: “eliminates the domain-specific information from the target domain”. This is a very strong claim which does not seem to be backed by evidence. Performing “domain alignment” is not the same as “eliminating” domain-specific information. Further, as we can see from Fig. 3, the sub-prototypes seem to be retaining domain-specific information.

* There are no sensitivity analyses for the several loss-balancing hyperparameters $\lambda_1, \lambda_2, \lambda_3$ (not even in the Supplementary). While the paper claims to have borrowed them from DCC, this approach is vastly different from DCC, and we need to check for sensitivity to these hyperparameters. Further, DCC does not have a reconstruction loss, so it is unclear how that hyperparameter is selected.

* There is no ablation study for the adaptive threshold $\lambda$. It should be compared to various fixed thresholds and the value of the adaptive threshold should also be plotted over the course of training to obtain more insights into its working.

* Other UniDA works, like OVANet [40] and [W1], study the sensitivity of their methods to the degree of openness (i.e. the number of shared/private classes) which changes the difficulty of the UniDA problem. This analysis is missing in this paper. This should be shown for a better understanding of the capabilities of the proposed method.

* Some more related work [W3-W4] on Open-Set DA and UniDA (apart from [W1, W2]) that is not discussed in this paper.

* Minor problems (typos):
    * L53: “adaption” → “adaptation”
    * L59: “shifts” → “shift”
    * L92: use `unknown’ i.e. use a backquote in LaTeX for it to properly render the opened and closed quotes like in L102.
    * L119: use math-mode for K in top-$K$.
    * L124: “varies” → “vary”
    * L126, 179: add space between text and \cite{...}
    * L134: “differenciates $\hat{Z}$ with” → “differentiates $\hat{Z}$ from”
    * L151: “max” → “maximum”
    * L166: “only the $K$” → “only the top-$K$”
    * L181: “$max$” → “$\max$”
    * L244: “fellow” → “following”

* Minor problems (grammatical errors):
    * L32: “aims” → “aiming”
    * L40: “Since such kind” → “Since this type”
    * L41: “almost happens in all the” → “occurs in almost all of the”
    * L59: “embedding give into” → “embedding is passed to”
    * L125: “sometimes is” → “is sometimes”

### References

[W1] Kundu et al., “Subsidiary Prototype Alignment for Universal Domain Adaptation”, NeurIPS22

[W2] Liu et al., “PSDC: A Prototype-Based Shared-Dummy Classifier Model for Open-Set Domain Adaptation”, IEEE Transactions on Cybernetics, Dec. 2022

[W3] Chen et al., “Evidential Neighborhood Contrastive Learning for Universal Domain Adaptation”, AAAI22

[W4] Garg et al., “Domain Adaptation under Open Set Label Shift”, NeurIPS22

**Questions:**

Please see the weaknesses section.

Overall, the technical contributions seem to be novel and intuitive. However, there are significant concerns regarding missing discussions on highly relevant work [W1], lack of mathematical rigor, missing sensitivity analyses and ablation studies, and the restrictiveness of requiring a CLIP-pretrained backbone. Hence, my rating is “4: borderline reject” at this time but I am willing to update my rating based on the rebuttal and discussion.

**Limitations:**

I appreciate that the paper provides both limitations and broader societal impact discussions in the Supplementary.

---

> ### Author Rebuttal · Authors · 2023-08-08
>
> ##### Re_Q1:
>
> Although the concept of the prototype is mentioned in [W1] and [W2], there are clear differences between theirs and our MemSPM.
>
> First, the meaning of prototype is different between [W1] and ours. In the [W1], the subsidiary prototype is extracted from randomly cropped images, which means the subsidiary prototypes only represent the low-level, morphological, and partial features of the image. These subsidiary prototypes don’t have complete semantic knowledge, and the method can’t learn the concept shift in the same category. Moreover, they still used the labeled category directly for alignment and adaptation. These prototypes can’t represent some part of the samples in one category.
>
> In contrast, our MemSPM allows memory items to extract complete semantic knowledge and maintain domain-invariant knowledge. To accomplish this, we use input-oriented embedding, which involves comparing the entire image feature with memory items. The memory can then sample a task-oriented embedding that represents the semantic knowledge of the input-oriented embedding. Our approach is designed to obtain a domain-invariant and semantic feature for categories with significant domain shifts. As a result, each sub-prototype can represent a sub-class in one category.
>
> Second, the purpose of [W2] is very different from our MemSPM. They aim to learn differences among unknown classes, which is like the DCC. It still extracts features and aligns the class across different domains directly based on one-hot labels, and doesn’t concern with the concept shift and difference in one category. However, our method can mine the sub-classes in one category when there exist significant concept shifts, reflecting the inherent differences among samples annotated as the same category. This helps universal adaptation with a more fine-grained alignment.
>
> ***
>
> ##### Re_Q2:
>
> Thanks for the advice. We have revised these errors for clearer clarification.
>
> **eq1**  We apply tensor operation using the Einstein summation notation on $W$ and $M$: $' nd, ckd->nkc'$. The memory shape is $[C * K * D]$ and n is batch size.
>
> **eq5**
> The $i$ dimension of $w$ is used for top-k.
> ***
>
> ##### Re_Q3:
>
> We have conducted experiments that adopt the backbone with ImageNet initialization. The performance of MemSPM on Officehome using **ViT-B/16(ImageNet)**  is **76.7%** (H-score), which is **7.5%** lower than MemSPM using **ViT-B/16(CLIP)**. Thus, adopting a better pre-trained encoder will result in better performance. We have tried the approach of retrieving sub-prototypes after a few epochs of regular training, but it cannot resolve the problem because it only reaches **64.3%** and the loss does not decrease in the following epochs.
>
> | Method | Backbone Pretrain | Ar2Cl | Ar2Pr | Ar2Rw | Cl2Ar | Cl2Pr | Cl2Rw | Pr2Ar | Pr2Cl | Pr2Rw | Rw2Ar | Rw2Cl | Rw2Pr | Avg  |
> |:-------:|:-------:|:-------:|:-------:|:-------:|:-------:|:-------:|:-------:|:-------:|:-------:|:-------:|:-------:|:-------:|:-------:|:------:|
> | DCC+MemSPM |ViT-B/16 ImageNet |  57.1    |   85.0    |   88.4    |   60.8   |    61.1   |   85.2    |   **83.5**    |   **76.1**    |  87.5     |  **82.7**    |  **77.3**     |  76.4   | 76.7 |
> | DCC+MemSPM |ViT-B/16 CLIP |   **78.1**    |   **90.3**    |   **90.7**    |   **81.9**   |    **90.5**   |   **88.3**    |   79.2    |   77.4    |  **87.8**     |   78.8   |  76.2     |  **91.6**   | **84.2** |
> | DCC+MemSPM Without Lcdd |ViT-B/16 CLIP|  75.9 |   75.4    |    86.4   |   80.1    |   71.6    |    87.5   |    70.1   |   87.1    |  88.7     |   74.2    |   73.5    |  88.8    | 79.8|
> | Fixed Threshold=0.005 DCC+MemSPM |ViT-B/16 CLIP |   64.6    |   86.7    |   87.4    |   63.3   |    68.5   |   79.3    |   65.9    |   65.8    |  81.4     |   70.7  |  68.8    |  85.5   | 73.9 |
>
> ***
>
> ##### Re_Q4:
>
> Thanks for your advice. We have modified the claim in this section. What we mean is that the sub-prototypes are all learned from the source domain, and the target input will be represented by these sub-prototypes. Therefore, we find that the task-oriented information retrieved from memory will mainly have features from the source domain. After that, the classifier can accurately classify, similar to how it does in the source domain. In **Figure 3**, the visualization seems to be from one domain, because our memory only has the source domain feature and the decoder was trained on it.
>
> ***
>
> ##### Re_Q5:
> Thanks for the advice. We have conducted experiments on the contribution of each loss. The L_ce is for the classification that can not be removed. The L_cdd is removed and the model achieves **79.8\%**, which is **4.4\%** lower than using the full loss function. Through experiment, we find that the coefficient of L_ce **($\lambda_{1} = 0.1$)** and the coefficient of L_cdd **($\lambda_{2} = 3$)** achieves the best performance. The reconstruction task loss (L_rec) has a slight improvement on the performance but is mainly designed for better visualizing and understanding the learned sub-prototypes. We will add these results to the revised manuscript.
>
> ***
>
> ##### Re_Q6:
> We find a best-performed fixed threshold of **0.005** through experiments. It limits the memory to learn sub-prototypes, which only achieved **73.9%** (H-score) on Officehome. Moreover, using the fixed threshold will add another hyperparameter to the MemSPM, which must be adjusted to different settings. We will add this to the revised manuscript.
> ***
>
> ##### Re_Q7: Openness of setting:
>
> The setting of different openness was listed in **Table 1**, and the results were listed in **Tables 2, 3, 4, and 5**. We also have done an additional setting on Officehome.
> | Unknown class   | Avg   |
> |:-------:|:-------:|
> | 5 | 82.3 |
> | 10 | 81.7 |
> | 50 | 84.2 |
>
> ***
>
> ##### Re_Q8: Related works:
>
> Thank you for your advice. We will add these works to the revised manuscript.
>
> ***
>
> ##### Minor problems (typos and grammatical errors):
>
> Thank you for your advice. We have revised these problems.

---

> > ### Comment · Reviewer_EAMn · 2023-08-18
> > **Response to Rebuttal**
> >
> > I thank the authors for their detailed response and appreciate their efforts in the process.
> >
> > Most of my concerns have been addressed and I have increased my rating.
> >
> > However, I advise the authors to also include the DCC performance with CLIP initialization (which is 74.4% as per the main paper) along with the performance of DCC+MemSPM with ImageNet initialization (which is 76.7% as per the rebuttal). This actually strengthens the rebuttal because it shows that MemSPM gives an improvement over DCC+CLIP even with ImageNet initialization. For future drafts, please add the performance of DCC with ImageNet initialization for ViT-B/16 model.

---

> > > ### Author Response · Authors · 2023-08-18
> > >
> > > Thank you for providing valuable feedback on our work. We have added the performance of DCC with CLIP initialization along with the performance of DCC+MemSPM with ImageNet initialization. We conduct experiments on DCC with ImageNet initialization for the ViT-B/16 model and will revise it in the final version.

---

### Official Review · Reviewer_YkYx · 2023-07-06

**Soundness:** 3 good
**Presentation:** 3 good
**Contribution:** 2 fair
**Rating:** 5
**Confidence:** 5

**Summary:**

This paper proposes a Memory-Assisted Sub-Prototype Mining (MemSPM) method that can learn the differences between samples belonging to the same category and mine sub-classes when there exists significant concept shift between them.

**Strengths:**

The writing of the article is very good. Graphical expressions such as t-SNE are very clear. The method have achieved relatively high classification H-score.

**Weaknesses:**

Some training details need to be explained, such as the selection of hyperparameters. How to adjust the N, S and lambda, and what criteria are based on? If it is based on the final experimental effect, it also indirectly depends on the label information of the target domain.
The scalability of the method is relatively poor. If the data set is large and there are many categories, will there be many prototypes required, and how will the method perform? It is crucial to have the Domainnet dataset in the experiments.

**Questions:**

mainly of the weaknesses.

**Limitations:**

This paper has no limitation sections.

---

> ### Author Rebuttal · Authors · 2023-08-08
>
> ##### Q1:
>
> Some training details need to be explained, such as the selection of hyperparameters. How to adjust the N, S, and lambda, and what criteria are based on? If it is based on the final experimental effect, it also indirectly depends on the label information of the target domain.
>
> ##### Re_Q1:
>
> We carefully conducted ablation experiments on these hyperparameters, which can be seen in **Figure 3(b)**. We find that MemSPM is insensitive to N, while larger S values result in better performance. As we are mainly concerned about the hyperparameters of memory structure, we find that we only need to set large S and N, and this setting depends solely on the GPU memory. Moreover, we consistently applied the **$\lambda_{1} = 0.1$** and the **$\lambda_{2} = 3$** on all datasets. Therefore, setting the hyperparameters does not require any information about the target domain and we can use the fixed hyperparameters in all datasets.
>
> ***
>
> ##### Q2:
>
> The scalability of the method is relatively poor. If the data set is large and there are many categories, will there be many prototypes required, and how will the method perform? It is crucial to have the Domainnet dataset in the experiments.
>
> ##### Re_Q2:
>
> We do have the results for **Domainnet**, VisDA, Office-31, and Office-Home presented in **Tables 2** and **3**. Domainnet, VisDA, and Officehome all require many prototypes to represent sub-classes, and MemSPM has demonstrated state-of-the-art performance on all these benchmarks. As mentioned in **Re_Q1**, the MemSPM only needs to set large values for N and S, which are solely related to GPU memory, when dealing with a large number of categories.

---

### Official Review · Reviewer_S9DQ · 2023-07-08

**Soundness:** 2 fair
**Presentation:** 3 good
**Contribution:** 2 fair
**Rating:** 6
**Confidence:** 5

**Summary:**

This work addresses the problem of universal domain adaptation by focusing on the intra-class structure within categories, which is often overlooked by existing methods.

The main contribution is the proposed Memory-Assisted Sub-Prototype Mining (MemSPM) method, which learns the differences between samples belonging to the same category and mines sub-classes in the presence of significant concept shift. By doing so, the model achieves a more reasonable feature space that enhances transferability and reflects inherent differences among samples.

Experimental evaluation demonstrates the effectiveness of MemSPM in various scenarios, achieving state-of-the-art performance on four benchmarks in most cases.

**Strengths:**

S1 : The primary contribution of this work is the introduction of sub-prototypes, learned from samples within the same category but exhibiting significant concept shift.   The utilization of sub-prototypes allows for a more fine-grained adaptation process, which is an intuitive and an interesting idea.  The ablation experiment Figure 3 (graph), supports the notion that mining sub-prototypes is indeed advantageous, as increasing the number of sub-prototypes (S) leads to a substantial performance improvement, from approximately 62% (with one sub-prototype per category) to around 80% (with 40 sub-prototypes per category).

S2: The results presented in Table 2 and Table 3 demonstrate significant performance improvements compared to previous works, with increases of +4.5% and +6.4% in H-score on DomainNet and Office-31 datasets for UniDA scenario. Additionally, there is a +1.6% improvement in H-score on the Office-Home dataset. It should be noted that the comparisons are not entirely apples-to-apples, as discussed in the weaknesses section.

**Weaknesses:**

W1: The utilization of CLIP-based embedding as mentioned in line 126 offers semantic capabilities that generalize across various domains (as shown by works such as [1, 2, ..] that build on top of CLIP). However, the importance of using CLIP-based embedding is not clearly demonstrated in the ablation analysis. A comparison between CLIP-based embedding, learned embedding (without pre-training), and ViT-B/16 (pre-trained on ImageNet) would provide valuable insights. Additionally, the lack of utilization of CLIP's semantic capabilities in prior works raises concerns about the apples-to-apples comparison of the results presented in Table 2 and Table 3.

W2: From the experiment section, the impact of different losses, such as cross-entropy (L_ce), domain alignment loss (L_cdd), and auxiliary reconstruction task (L_rec), on model performance is not clearly explained in the experiment section. Understanding the contribution of each loss would enhance the understanding of the paper.

W3: The sensitivity of hyperparameters across different scenarios, such as Open-Set Domain Adaptation (OSDA) and UniDA, is not adequately addressed in this section. Investigating the sensitivity of hyperparameters would provide valuable insights into their impact on model performance.

W4: Section 3.3.3 discusses the "Adaptive Threshold Technique for More Efficient Memory," but there is a lack of experimental details showcasing the memory efficiency of this technique. Without such evidence, it becomes challenging to fully appreciate the technical contribution.

W5: While the motivation and the main idea of mining sub-prototypes are novel, it is worth noting that memory-based prototype mining was explored earlier in works like [3]. This observation slightly diminishes the overall technical contribution..

W6: Supplementary material Figure 1 reveals that a significant portion (>60%) of the sub-prototype visualizations are not interpretable. This undermines the contribution of interpretability in this work.
[1] Rinon Gal and Or Patashnik and Haggai Maron and Gal Chechik and Daniel Cohen-Or StyleGAN-NADA: CLIP-Guided Domain Adaptation of Image Generators, ACM Transactions on Graphics
[2] Boyi Li, Kilian Q. Weinberger, Serge Belongie, Vladlen Koltun, René Ranftl, Language-driven Semantic Segmentation, ICLR 2022
[3]Tarun Kalluri , Astuti Sharma, Manmohan Chandraker.\ MemSAC: Memory Augmented Sample Consistency for Large Scale Domain Adaptation, ECCV 2022

**Questions:**

Please refer the weaknesses section for the related questions that need more clarification.

**Limitations:**

A notable limitation of the study is the lack of clarity regarding the contribution of various components of the proposed method to the overall performance. Specifically, the impact of CLIP-based embedding, which has demonstrated generalizable capabilities even in zero-shot scenarios across domains, needs to be thoroughly understood to fully appreciate the proposed components. Gaining insights into the individual contributions of different components would provide a deeper understanding of their influence on the overall performance. Further investigations or additional analyses focusing on these aspects would enhance the comprehensiveness and rigor of the study.

---

> ### Author Rebuttal · Authors · 2023-08-08
>
> Thanks for supporting our work.
>
> ##### Re_W1:
>
> Although CLIP-based embedding does have some cross-domain knowledge, it still cannot address the large domain gap that existed in the benchmarks. The baseline that simply adopts the CLIP encoder has been tested on the Officehome dataset only achieving **77.2\%** (H-score), which is **7.0\%** lower than our MemSPM.
>
> As you suggested, we have conducted experiments to compare ViT-B/16 (pre-trained by CLIP), ViT-B/16 (pre-trained on ImageNet), and ViT-B/16 (without pre-training). The performance of MemSPM on Officehome using ViT-B/16 (ImageNet)  is **76.7\%** (H-score), which is **7.5\%** lower than MemSPM using ViT-B/16 (pre-trained on CLIP). Additionally, the ViT-B/16 (without pre-training) only achieves **64.3\%**, which is **19.9\%** lower than that using ViT-B/16 (pre-trained on CLIP). These experiments demonstrate that a better pre-trained encoder can benefit sub-prototype mining.
>
> In **Table 2** and **Table 3**, all the methods of GLC, DCC, and MemSPM adopt the ViT-B/16 backbone (pre-trained by CLIP), so the comparison is fair. The GLC is the SOTA method of UniDA and DCC is the method we based on.
>
> | Method | Backbone Pretrain | Ar2Cl | Ar2Pr | Ar2Rw | Cl2Ar | Cl2Pr | Cl2Rw | Pr2Ar | Pr2Cl | Pr2Rw | Rw2Ar | Rw2Cl | Rw2Pr | Avg  |
> |:-------:|:-------:|:-------:|:-------:|:-------:|:-------:|:-------:|:-------:|:-------:|:-------:|:-------:|:-------:|:-------:|:-------:|:------:|
> | DCC+MemSPM |ViT-B/16 None |   50.7    |   78.4    |   85.6    |   50.2   |    60.7   |   67.1    |   58.2    |   44.1    |  77.9     |   67.1  |  50.3    |  81.7   | 64.3 |
> | DCC+MemSPM |ViT-B/16 ImageNet |  57.1    |   85.0    |   88.4    |   60.8   |    61.1   |   85.2    |   **83.5**    |   **76.1**    |  87.5     |  **82.7**    |  **77.3**     |  76.4   | 76.7 |
> | DCC+MemSPM |ViT-B/16 CLIP |   **78.1**    |   **90.3**    |   **90.7**    |   **81.9**   |    **90.5**   |   **88.3**    |   79.2    |   77.4    |  **87.8**     |   78.8   |  76.2     |  **91.6**   | **84.2** |
> | DCC+MemSPM Without Lcdd |ViT-B/16 CLIP|  75.9 |   75.4    |    86.4   |   80.1    |   71.6    |    87.5   |    70.1   |   87.1    |  88.7     |   74.2    |   73.5    |  88.8    | 79.8|
> | Fixed Threshold=0.005 DCC+MemSPM |ViT-B/16 CLIP |   64.6    |   86.7    |   87.4    |   63.3   |    68.5   |   79.3    |   65.9    |   65.8    |  81.4     |   70.7  |  68.8    |  85.5   | 73.9 |
> ***
>
> ##### Re_W2:
> Thanks for the advice. We have conducted experiments on the contribution of each loss. The L_ce is used for the classification that cannot be removed. The L_cdd is removed and the model achieves **79.8\%**, which is **4.4\%** lower than using the full loss function. In our early experiment, we observe that the coefficient of L_ce **($\lambda_{1} = 0.1$)** and the coefficient of L_cdd **($\lambda_{2} = 3$)** achieve the best performance.
> The reconstruction task loss (L_rec) has a slight performance improvement but is mainly designed for better visualizing and understanding the learned sub-prototypes. Since we focused on studying the hyperparameters of the proposed memory structure, the ablation results of loss functions have not been presented in the paper due to the space limit. We will add them to the revised manuscript.
>
>
> ***
>
> ##### Re_W3:
>
> We examined the impact of these hyperparameters on MemSPM and the results are illustrated in **Figure 3(b)**. In different scenarios, we used the same hyperparameters of memory structure. MemSPM reaches the SOTA results on OSDA and UniDA in **Tables 2 and 3**, and comparable results on PDA in **Table 4**, which means the hyperparameters of MemSPM are not sensitive to different scenarios.
>
> ***
>
> ##### Re_W4:
>
> Thank you for your advice. Our memory structure is randomly initialized. We find a best-performed fixed threshold of **0.005** through experiments. It limits the memory to learn sub-prototypes, which only achieved **73.9\%** (H-score) on Officehome. Moreover, using the fixed threshold will add another hyperparameter to the MemSPM, which must be adjusted to different settings. We will add more details to the revised manuscript.
>
> ***
>
> ##### Re_W5:
>
> There are several major differences between MemSAC [1] and our proposed MemSPM method.
>
> First, MemSAC employs a non-parametric memory bank that directly stores the features extracted by the encoder. This approach leaves too much domain-specific knowledge in the memory feature space. In contrast, our method employs a learnable memory that samples memory items based on input-oriented embedding. This creates a task-oriented embedding with less domain-specific knowledge.
>
> Second, the memory structure used in MemSAC can be traced back to the work published in [2]. So, we view this structure as analogous to widely-used structures such as Transformer, and ResNet. Our use of the learnable memory for task-oriented feature mining, as well as our improvement in the memory structure for sub-prototype mining, represents a significant departure from previous work.
>
> ***
>
> ##### Re_W6:
>
> In our supplementary material, we show the memory visualization results without dropping imperfect ones.  For the robustness and scalability of the MemSPM, we set large N and S that can fit all datasets. We also conducted ablation experiments on N and S, which can be seen in ****Figure 3(b)****. We find that MemSPM is insensitive to N, while a larger S value results in better performance. So, it is a fact that our learned memory bank contains redundant items to some extent, meaning the number of memory items is more than necessary for the sub-prototypes. As a result, some items do not look that good semantically. It doesn't affect the interpretability of our method.
>
> ***
>
> [1] Tarun Kalluri, et al.  MemSAC: Memory Augmented Sample Consistency for Large Scale Domain Adaptation. ECCV 2022
>
> [2] Sukhbaatar, Sainbayar, Jason Weston, and Rob Fergus. End-to-end memory networks. NeurIPS 2015.

---

> > ### Comment · Reviewer_S9DQ · 2023-08-19
> > **Response to Rebuttal**
> >
> > I thank the authors for their comprehensive response and value their effort in the process.
> >
> > They have adequately addressed the majority of my concerns, leading to an adjustment in my rating.

---

### Author Rebuttal · Authors · 2023-08-10

We thank all the reviewers for their insightful feedback and list some responses to common questions in this section.
***
##### Q1: Impact of CLIP:
CLIP-based embedding does have some cross domains knowledge but it was still influenced by the larger domain gap. The simple baseline of CLIP has been tested in officehome only achieving **77.2\%** (H-score), which still is **7.0\%** lower than our MemSPM.

We also have conducted experiments to compare ViT-B/16 (pre-trained by CLIP), ViT-B/16 (pre-trained on ImageNet), and ViT-B/16 (Without pre-trained). The performance of MemSPM in officehome using ViT-B/16 (ImageNet)  is **76.7\%** (H-score) which is **7.5\%** lower than MemSPM using ViT-B/16 (pre-trained on CLIP). Additionally, the ViT-B/16 (Without pre-trained) only achieves **64.3\%**, which is **19.9\%** lower than that using ViT-B/16 (pre-trained on CLIP). These experiments demonstrate that a better pre-trained encoder can benefit sub-prototype mining.

| Method | Backbone Pretrain | Ar2Cl | Ar2Pr | Ar2Rw | Cl2Ar | Cl2Pr | Cl2Rw | Pr2Ar | Pr2Cl | Pr2Rw | Rw2Ar | Rw2Cl | Rw2Pr | Avg  |
|:-------:|:-------:|:-------:|:-------:|:-------:|:-------:|:-------:|:-------:|:-------:|:-------:|:-------:|:-------:|:-------:|:-------:|:------:|
| CLIP-Baseline |ViT-B/16 CLIP|  64.6 |   84.3    |    78.1   |   73.7    |   88.2    |    86.5   |    68.1   |   68.7    |  **89.6**     |   68.5    |   69.4    |  86.6    | 77.2 |
| DCC+MemSPM |ViT-B/16 None |   50.7    |   78.4    |   85.6    |   50.2   |    60.7   |   67.1    |   58.2    |   44.1    |  77.9     |   67.1  |  50.3    |  81.7   | 64.3 |
| DCC+MemSPM |ViT-B/16 ImageNet |  57.1    |   85.0    |   88.4    |   60.8   |    61.1   |   85.2    |   **83.5**    |   **76.1**    |  87.5     |  **82.7**    |  **77.3**     |  76.4   | 76.7 |
| DCC+MemSPM |ViT-B/16 CLIP |   **78.1**    |   **90.3**    |   **90.7**    |   **81.9**   |    **90.5**   |   **88.3**    |   79.2    |   77.4    |  87.8     |   78.8   |  76.2     |  **91.6**   | **84.2** |

***

##### Q2: Hyperparameters and Loss Function:
We conducted ablation experiments on hyperparameters, the results of which can be seen in **Figure 3(b)**. We found that MemSPM was insensitive to N, while larger S values resulted in better performance. For these experiments, we discovered that we only need to set a large S and N, and this setting depends solely on the GPU memory.

We have conducted experiments on the contribution of each loss. The L_ce is for the classification that can not be removed. The L_cdd is removed and the model achieves **79.8%**, which is **4.4%** lower than using the full loss function. After the experiment, we found that the coefficient of L_ce **($\lambda_{1} = 0.1$)** and the coefficient of L_cdd **($\lambda_{2} = 3$)** which were provided by DCC had the best performance.

For reconstruction task loss (L_rec), it wasn’t a main part of our loss. And in our experiment, we found that L_rec did a very small impact on the performance, which is just for visualization. As we are mainly concerned with hyperparameters of memory structure, these ablation results of loss had no space to list in. For hyperparameters of the loss function, we fixed them for all datasets. Therefore, setting the hyperparameters does not require hyperparameters turning and any information about the target domain. We will add more details in the revised manuscript.

| Method | Backbone Pretrain | Ar2Cl | Ar2Pr | Ar2Rw | Cl2Ar | Cl2Pr | Cl2Rw | Pr2Ar | Pr2Cl | Pr2Rw | Rw2Ar | Rw2Cl | Rw2Pr | Avg  |
|:-------:|:-------:|:-------:|:-------:|:-------:|:-------:|:-------:|:-------:|:-------:|:-------:|:-------:|:-------:|:-------:|:-------:|:------:|
| DCC+MemSPM Without Lcdd |ViT-B/16 CLIP|  75.9 |   75.4    |    86.4   |   80.1    |   71.6    |    87.5   |    70.1   |   **87.1**    |  **88.7**     |   74.2    |   **88.8**    |  73.5    | 79.8|
| DCC+MemSPM |ViT-B/16 CLIP |   **78.1**    |   **90.3**    |   **90.7**    |   **81.9**   |    **90.5**   |   **88.3**    |   **79.2**    |   77.4    |  87.8     |   **78.8**   |  76.2     |  **91.6**   | **84.2** |

***

##### Q3: Effectiveness of Adaptive Threshold
We find a best-performed fixed threshold of **0.005** through experiments. It limits the memory to learn sub-prototypes, which only achieved **73.9%** (H-score) on Officehome. Moreover, using the fixed threshold will add another hyperparameter to the MemSPM, which must be adjusted to different settings. We will add this to the revised manuscript.

| Method | Backbone Pretrain | Ar2Cl | Ar2Pr | Ar2Rw | Cl2Ar | Cl2Pr | Cl2Rw | Pr2Ar | Pr2Cl | Pr2Rw | Rw2Ar | Rw2Cl | Rw2Pr | Avg  |
|:-------:|:-------:|:-------:|:-------:|:-------:|:-------:|:-------:|:-------:|:-------:|:-------:|:-------:|:-------:|:-------:|:-------:|:------:|
| DCC+MemSPM |ViT-B/16 CLIP |   **78.1**    |   **90.3**    |   **90.7**    |   **81.9**   |    **90.5**   |   **88.3**    |   **79.2**    |   **77.4**    |  **87.8**     |   **78.8**   |  **76.2**     |  **91.6**   | **84.2** |
| Fixed Threshold=0.005 DCC+MemSPM |ViT-B/16 CLIP |   64.6    |   86.7    |   87.4    |   63.3   |    68.5   |   79.3    |   65.9    |   65.8    |  81.4     |   70.7  |  68.8    |  85.5   | 73.9 |

---

> ### Author Response · Authors · 2023-08-18
> **New Table of Rebuttal Results**
>
> According to the advice of the response, we add the performance of DCC with CLIP initialization along with the performance of DCC+MemSPM with ImageNet initialization.
>
> ***
> | Method | Backbone Pretrain | Ar2Cl | Ar2Pr | Ar2Rw | Cl2Ar | Cl2Pr | Cl2Rw | Pr2Ar | Pr2Cl | Pr2Rw | Rw2Ar | Rw2Cl | Rw2Pr | Avg  |
> |:-------:|:-------:|:-------:|:-------:|:-------:|:-------:|:-------:|:-------:|:-------:|:-------:|:-------:|:-------:|:-------:|:-------:|:------:|
> | DCC+MemSPM |ViT-B/16 ImageNet |  57.1    |   85.0    |   **88.4**    |   60.8   |    61.1   |   **85.2**    |   **83.5**    |   **76.1**    |  **87.5**     |  **82.7**    |  **77.3**     |  76.4   | **76.7** |
> | DCC |ViT-B/16 CLIP |   **62.6**    |  **88.7**    |   87.4    |  **63.3**   |    **68.5**  |   79.3    |   67.9    |   63.8    |  82.4     |   70.7  |  69.8    |  **87.5**   | 74.4 |
>
> ***
> | Method | Backbone Pretrain | Ar2Cl | Ar2Pr | Ar2Rw | Cl2Ar | Cl2Pr | Cl2Rw | Pr2Ar | Pr2Cl | Pr2Rw | Rw2Ar | Rw2Cl | Rw2Pr | Avg  |
> |:-------:|:-------:|:-------:|:-------:|:-------:|:-------:|:-------:|:-------:|:-------:|:-------:|:-------:|:-------:|:-------:|:-------:|:------:|
> | CLIP-Baseline |ViT-B/16 CLIP|  64.6 |   84.3    |    78.1   |   73.7    |   88.2    |    86.5   |    68.1   |   68.7    |  **89.6**     |   68.5    |   69.4    |  86.6    | 77.2 |
> | DCC+MemSPM |ViT-B/16 None |   50.7    |   78.4    |   85.6    |   50.2   |    60.7   |   67.1    |   58.2    |   44.1    |  77.9     |   67.1  |  50.3    |  81.7   | 64.3 |
> | DCC+MemSPM |ViT-B/16 ImageNet |  57.1    |   85.0    |   88.4    |   60.8   |    61.1   |   85.2    |   **83.5**    |   **76.1**    |  87.5     |  **82.7**    |  **77.3**     |  76.4   | 76.7 |
> | DCC+MemSPM |ViT-B/16 CLIP |   **78.1**    |   **90.3**    |   **90.7**    |   **81.9**   |    **90.5**   |   **88.3**    |   79.2    |   77.4    |  87.8     |   78.8   |  76.2     |  **91.6**   | **84.2** |
> | DCC |ViT-B/16 CLIP |   62.6    |   88.7    |   87.4    |   63.3   |    68.5   |   79.3    |   67.9    |   63.8    |  82.4     |   70.7  |  69.8    |  87.5   | 74.4 |

---

> ### Comment · Area_Chair_epsk · 2023-08-18
>
> Thank the authors for the rebuttal. PCs and I have reminded the reviewers to respond to the rebuttals as soon as possible. The final decision will depend on both the reviews and rebuttal into account.
>
> @Reviewers: This message is yet another reminder. Please try to respond to the rebuttal asap.
>
> --AC

---

### Decision · Program_Chairs · 2023-09-21

**Decision:**

Reject

**Comment:**

Five experts reviewed the paper. Reviewers S9DQ, EAMn, and QTwQ liked the concept of prototypes and the results. The rebuttal addressed their questions well, so they raised their ratings in the rebuttal phase. Reviewer YkYx ranked the paper around the borderline. However, Reviewer QTwQ shared with AC that the rebuttal did not convince them. In particular, QTwQ's first three questions/concerns remained. AC read the paper in addition to the reviews and rebuttal and agreed with Reviewer QTwQ's comments that the sub-prototypes' contribution to the overall method was unclear and that the hyperparameters desired further study. Moreover, no reviewer was especially excited about the work, though three reviewers raised ratings after the rebuttal. Hence, the decision is not to recommend the paper for acceptance. The paper clearly has merits, and we hope the reviews will be helpful for the authors to improve the paper for a submission somewhere else.